# Willingness to trust is reduced by loneliness and paranoia
Gabriele Bellucci [1,2] ✉, Mehdi Keramati[3], Esther Hanssen[4] & Anne-Kathrin Fett [3,5] ✉

Loneliness is associated with negative social behaviors, impairing social relationships. However, the underlying mechanisms are poorly understood. Here, we investigated the relationship between paranoid thoughts and lonely individuals' willingness to rely on expectations of partner reciprocity in an investment game with individuals with and without psychosis (54 participants). We found that loneliness and paranoia were strongly correlated with each other and with more distrustful behavior after breaches of trust. Sensitivity to changes in partner reciprocity was higher in lonelier and more paranoid individuals. Lonelier individuals also trusted highly reciprocating partners less. Computational modeling revealed that lonelier and more paranoid individuals were less willing to rely on expectations of partner reciprocity. Importantly, these effects were observed in both patients and controls, indicating the important role of loneliness and paranoia in both clinical and general populations. These findings demonstrate how loneliness relates to social behaviors and expectations, pointing to important downstream implications for lonely individuals' relationships.

Loneliness is defined as the feeling of perceived social isolation triggered by unsatisfying relationships in terms of quality and quantity[1]. It is associated with poorer physical and mental health, causing burden and suffering to lonely individuals[2–4]. Importantly, loneliness is thought to lead to social withdrawal and reduced motivation for social connection by increasing sensitivity to social threats[5,6]. Social isolation and a perceived lack of social support are common in mental disorders and social withdrawal has been recognized as a transdiagnostic symptom of the most frequent psychiatric disorders[7,8].

Aspects of social disconnection are pronounced in individuals with psychotic disorders, and loneliness has been linked to central symptoms of psychosis, including social threat perception and anhedonia (i.e., lack of reward/pleasure from social interactions)[9]. However, previous findings indicate small-to-moderate associations between loneliness and psychotic symptoms[4,10], implying that while these constructs are related––particularly through overlapping social-cognitive vulnerability––they remain distinct in etiology and expression. Most available studies are cross-sectional and focus on symptom correlations rather than how loneliness shapes real-world social behaviors. Examining how loneliness modulates social interactions—such as influencing perceptions of trustworthiness, expectations of reciprocity, or responsiveness to social cues—in individuals with and without psychotic disorders may help clarify transdiagnostic mechanisms of social dysfunction and address a critical gap in the literature[10,11].

Preliminary evidence suggests that loneliness is characterized by cognitive biases that lead to overly negative expectations of others[12]. A general negativity bias makes lonely individuals pay more attention to threatening interactions, be more aware of negative evaluations from others, remember more negative feedback, and form more negative expectations of others' social behaviors[13–16]. Some evidence indicates that a negative outlook on others is particularly more pronounced for close acquaintances than strangers[17]. These negative expectations and evaluations of others in lonely individuals have been suggested to foster paranoid thinking (i.e., the inflated belief that others have malevolent intentions)[18–20], which contribute to social isolation[21–23]. This may further exacerbate feelings of loneliness, as well as psychotic experiences and paranoid delusions[24–27]. For example, loneliness has been linked to paranoia via mediating cognitive-affective mechanisms[28] and suspiciousness towards others (i.e., paranoid thoughts) has been found to worsen over time following feelings of loneliness, with the duration of these feelings related to a decrease in trust in others[29,30]. As preliminary questionnaire-based evidence has also shown that reducing loneliness may lead to a decrease of paranoid symptoms[31], it is plausible that feelings of loneliness might promote more paranoid thoughts[29,32] in individuals who manifest higher levels of distrust[33–36] and are less likely to cooperate with others[37,38].

In particular, negative social expectations have been argued to contribute to a lack of interpersonal trust in lonely individuals[12]. As a

¹Center for Decision Sciences, Royal Holloway, University of London, Egham, UK. ²Department of Psychology, Royal Holloway, University of London, Egham, UK. ³Department of Psychology and Neuroscience, School of Health and Medical Sciences, City St George's, University of London, London, UK. ⁴Hersencentrum Mental Health Institute, Amsterdam, the Netherlands. ⁵Department of Psychosis Studies, Institute of Psychiatry, Psychology and Neuroscience, London, UK.
✉e-mail: gabriele.a.bellucci@gmail.com; anne-kathrin.fett@city.ac.uk

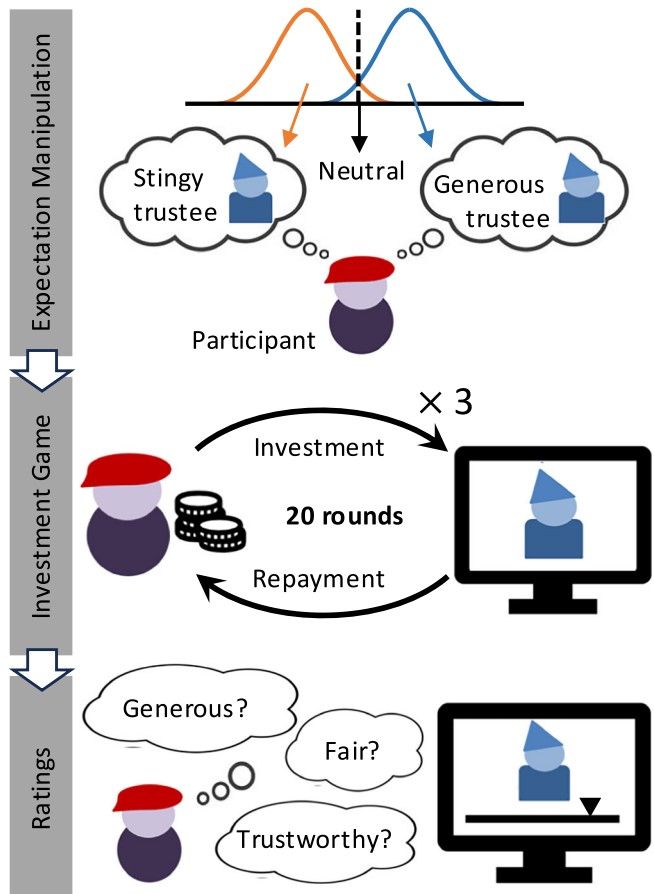

**Fig. 1 | Experimental design and modeling approach.** At the beginning of the investment game, participants' (investor) expectations of their partner's (trustee) returns were manipulated by making them believe that their partner was generous (positive condition) or stingy (negative condition); in the neutral condition, no expectation manipulation took place (within-subject condition). Participants played 20 trials and the amount they shared in the game (investment) measured their trialwise trust in each of the three partners (within subject condition). At the end of the game, they were asked to rate the trustworthiness, fairness, and generosity of each of the partners in that game (on a 7-point Likert scale).

social behavior that signals a person's attitudes towards others based on personal social expectations, trust is a particularly important social signal that can provide valuable insights into how loneliness and paranoia relate to reasoning about others during social interactions[39,40]. Recent work using the investment game––an economic paradigm that captures trust in reciprocity[41]––has shown that lonelier individuals have more positive attitudes towards new social partners and trust them to a greater extent in anonymous, single interactions[16,42]. At the same time, though, lonely individuals appear to attribute hostile intent[17,43] and malevolent intentions to others[19,20], showing paranoid thinking. These maladaptive cognitions have been related to failures of accurately adjusting trust in response to a partner's reciprocity[34,35].

Various mechanisms might explain these thinking patterns. For instance, previous work has suggested a biased social learning mechanism in lonely individuals where lonelier individuals integrate more negative than positive information about others' social behaviors[44]. This downweighing of positive information and beliefs about others could lead to similarly biased behavioral patterns. It is, thus, still unclear whether lonelier individuals form inaccurate social beliefs due to suboptimal learning or are simply reluctant to rely on positive beliefs about others[26]. For instance, their elevated sensitivity to being vulnerable to others might be associated with more cautious and suspicious behaviors in trusting interactions. Further, paranoid thinking, which has been associated with suspiciousness about others' good

intentions[18], might be associated with heightened perception of vulnerability and, hence, lower levels of trust.

Here, we investigated the relationships of loneliness and paranoia with belief formation and trust in a sample of participants with and without a psychotic disorder. Including both individuals with psychotic disorders and healthy controls allows us to examine whether the associations between loneliness, paranoia, and distrust are specific to clinical psychopathology or reflect broader, transdiagnostic mechanisms, given that these processes operate along a continuum in the general population[10,45]. Combining a modified investment game (Fig. 1) with Bayesian computational modeling, we investigated whether lonely individuals responded to their partner's reciprocity by showing a differential weighting of their expectations. We expect that if lonely individuals differentially weighted their expectations of partner reciprocity based on the observed behavioral feedback, they would show a different sensitivity to changes in partner reciprocity. For instance, they might be more cautious after signs of decreasing cooperation but more generous after signs of increasing cooperation. Further, we tested the relationship of paranoid thinking with expectation formation and weighting in lonely individuals. We hypothesized that higher levels of paranoid thoughts might bias lonely individuals' willingness to rely on their expectations of partner reciprocity due to higher perceived vulnerability, leading to a decrease in trust.

## Methods
### Participants
The sample included fifty-four participants (23-57 years; $M = 37.81$, $SD = 9.19$; 15 females, 39 males). Twenty-nine participants had a diagnosis of non-affective psychosis, and twenty-five participants were healthy matched controls without a personal or family history of psychosis. Our initial sample size considerations of $N = 50$ were based on the effects of a similar experimental manipulation of participants' expectations on trust in a sample of patients with schizophrenia and other psychotic disorders and a sample of healthy controls (see cooperative context of Table 3 in[35]), which yielded an effect size of $d = .71$ calculated for an independent sample t-test, with a desired power of 0.80. With a sample of $N = 54$, our study had 80% power to detect an effect size of $d = 0.78$. This is an effect size comparable to previous studies[35]. Inclusion criteria for all participants were: (i) age between 18 and 65 years; (ii) good understanding of the English language; (iii) IQ higher than 70; and (iv) a diagnosis according to the ICD-10 (for patients). We also had two additional exclusion criteria, such as a history of neurological conditions, and a diagnosis of alcohol/drug dependence. Recruitment of patients was completed via the Psychosis Studies department at the Institute of Psychiatry, Psychology, Neuroscience, King's College London. Controls were recruited through online advertisements and circular emails. The study was approved by the London-Harrow Research Ethics Committee [14/LO/0071], and all participants were provided written informed consent. This study was not preregistered.

### Task
To measure trust, a modified multi-round investment game was used (Fig. 1)[30,41]. In this neuroeconomic computer-based game, consisting of 20 trials each, participants played the role of the investor. They were instructed they play with three different human counterparts in another location via the Internet. In fact, they were playing against a computer that was pre-programmed to behave in a probabilistic and benevolent manner. Trustees were pre-programmed to play an overall benevolent, tit-for-tat strategy. The first repayment (i.e., reciprocity) was either 1.0, 1.1, 1.2, 1.3, 1.4, or 1.5 times (i.e., repayment rates) the first investment (i.e., the amount shared by the participant and indicative of their trust), with an equal chance for returns of each repayment rate. After the first trial, the repayment rates were updated depending on the amount being invested compared to the amount invested in the previous trial (i.e., trust change: $\delta_t = trust_t - trust_{t-1}$). When the investor's current trust increased or when participants continued to invest the maximum amount (£10), then an increment of 0.05 was added to each of the randomly selected repayment rates until each of the factors reached 1.6,

thus reflecting higher reciprocity. Each repayment rate decreased with the same increments (−0.05) when trust decreased or remained at the minimum (£0), with a minimum value of 1 for each repayment rate, thus reflecting lower reciprocity. This way of programming ensured reciprocity that would resemble subtle changes in trust. Moreover, since repayment rates were chosen probabilistically between the above range, they still allowed for trials in which trustees manifested either positive (increases of) or negative (decreases of) reciprocity (i.e., trustees were not deterministically mimicking participants' strategy). This design allowed us to investigate participants' reactions to positive and negative reciprocity.

The investment game had three conditions. In one, participants were provided with no information about the partner's initial reciprocity (neutral), while in the other two, they were led to believe that the partner was either benevolent (positive) or malevolent (negative). At the start of each trial, the initials of the other player were displayed on the computer screen, together with information on the partner's reciprocity in the positive and negative conditions. Participants (investors) received an initial endowment of £10 and indicated, on a horizontal scale from zero to ten pounds, how much they wanted to share with the other player. The chosen amount was tripled during the transaction, and after this, the partner gave the repayment. During the decision-making of the partners, participants saw the partner's initials on the screen with the message that they had to wait for a response. At the end of the trial, they saw their own total investments (kept and given amount) and total earnings for both players. After this, a new trial started. Due to technical issues, only one condition was collected for one participant.

At the end of the experiment, we asked participants to rate their partners' trustworthiness, fairness, and generosity on a 7-point Likert scale to obtain a measure of participants' overall impressions of their partners' social behavior during the investment game. Due to technical issues, ratings for one participant are missing.

## Experience sampling measures

To assess loneliness and paranoia, we used the experience sampling method (ESM). Participants completed an app-based ESM questionnaire, which included questions on loneliness and paranoia up to ten times a day following a reminder. Loneliness was assessed for every participant based on the average score of their responses over one week to the item "I feel lonely", as previously done[20] and currently recommended by the Office of National Statistics[46]. Paranoia was assessed for every participant based on the average score of their responses over one week to the following four items previously used[47]: "I feel suspicious"; "I feel safe (reverse scored)"; "I feel others dislike me"; and "I feel others intend to harm me". ESM responses did not differ between groups for either loneliness ($t_{52} = 0.85, p = .398$) or paranoia ($t_{52} = 1.05, p = .297$). ESM measures were used to assess loneliness and paranoia in the flow of daily life. ESM measures have the benefit that they are less vulnerable to many sources of error that are inherent in many traditional assessment techniques (e.g., memory and recall biases), yielding more reliable measures, particularly in patients who often suffer from cognitive problems[48–50]. Moreover, we also collected paranoia scores using the Green et al. Paranoid Thoughts Scale (GPTS)[51], as a well-established control measure of paranoia, to validate our ESM measure of paranoia. Our ESM measure of paranoia was highly correlated with GPTS score for both GPTS subscales (reference: $\rho_{52} = .62, p < .0001$; persecution: $\rho_{52} = .74, p < .0001$), indicating high validity of our ESM measure. As expected, GPTS scores from the persecution scale (reflecting paranoia) were significantly higher in the clinical sample ($t_{52} = 2.56, p = .014$). Moreover, like for the ESM measures of paranoia (controls: $SD = 0.98$; patients: $SD = 1.46$), GPTS scores had higher variability in our clinical sample (controls: $SD = 7.61$; patients: $SD = 18.41$). ESM scores for loneliness revealed a higher average for our clinical sample (controls: $M = 2.00$; patients: $M = 2.62$) but similar variability in scores (controls: $SD = 1.05$; patients: $SD = 0.98$). Importantly, these descriptive differences were relatively small and Welch's t-tests for unequal variance revealed virtually the same between-group results as parametric t-tests for ESM loneliness scores ($t_{52} = 2.24, p = .0297$), ESM paranoia scores ($t_{52} = 2.39, p = .0207$), and GPTS persecution scores

($t_{52} = 2.70, p = .010$). Questionnaires (signaled by a beep) occurred at pseudo-random moments with at least 15 min and at most 1.5 h between two consecutive questionnaires throughout the day––between 8.00 am and 10.30 pm. The ESM questionnaire was completed for seven successive days, and ESM items were rated on a seven-point Likert scale ranging from "not at all" (1) to "very" (7). Participants who owned an iPhone ran the app on their own device; all other participants were lent an iPod.

## Behavioral analyses

A mixed-effect Bayesian model was implemented to predict trialwise trust (amount shared by participants) with group, experimental condition, and trialwise reciprocity (amount shared back by the algorithm) as regressors. A similar model was implemente,d predicting trust changes (as the difference in investments between the current and the previous trial) and changes (increases/decreases) in partner reciprocity (as the difference in partner reciprocity between the last two trials) as regressors. To test for time effects over trials, a time regressor was further introduced to code for three categorical time periods (i.e., early, mid, and late). Similar Bayesian regression models with group and experimental conditions as regressors were run to predict post-game impression judgments (i.e., ratings of fairness, generosity, and trustworthiness on a 7-point Likert scale). Finally, subject-level Bayesian regression models with loneliness and paranoia as regressors were run to test the associations of loneliness and paranoia with model parameters. For all models, we used uninformative, Gaussian beta priors, i.e., $\beta \sim \mathcal{N}(0, 5)$. Stan was used to fit models with the no-U-turn sampler for efficient exploration of posterior estimates (https://mc-stan.org) with four chains, 1000 tuning steps, burn-in steps, and 20,000 samples. Visual inspection of the traces and the Gelman-Rubin statistics ($\hat{R}$) was used to assess convergence. Q-Q plots and histograms were used to visually inspect homoscedasticity and normality of residuals.

All models entailed demographic control variables including age, sex (0 = female; 1 = male), living condition (0=not alone; 1=alone), ethnicity (0 = other; 1 = white), and education (0 = none; 1 = primary; 2 = secondary; 3 = college; 4 = university). Results reveal that more educated participants ($\beta = 0.50$, 90% CI = [0.13, 0.88], *probability of direction*, *pd*, = 0.98) who were not living alone ($\beta = −0.86$, 90% CI = [−1.63, −0.02], $pd = 0.95$) trusted more in the investment game. There were no significant associations between trust and age ($\beta = 0.01$, 90% CI = [−0.04, 0.04], $pd = 0.53$) or sex ($\beta = 0.20$, 90% CI = [−0.57, 1.01], $pd = .68$). Finally, we found that participants living alone reported higher levels of loneliness ($\beta = 0.61$, 90% CI = [0.13, 1.12], $pd = 0.97$), while more educated participants ($\beta = −0.37$, 90% CI = [−0.68, −0.05], $pd = 0.97$) reported lower levels of paranoia. All analyses looking at associations between trust, loneliness, paranoia, and demographic characteristics were controlled for group status (patient vs. control).

## Computational analyses

We fit two classes of mathematical models to investigate the computational principles underlying participants' trust decisions (Tab. S1). The first class refers to a type of recently-proposed model known as vulnerability model, while the second class refers to a class of inequity-aversion models[52], which aims to capture how much individuals dislike unequal outcomes[53].

In the vulnerability model[54], willingness to trust is a subject-specific parameter that captures the degree to which one is willing to rely on their expectations of a partner's behavior against the danger of being vulnerable to the partner. In the investment game, this is the trade-off between the possibility of experiencing a breach of trust (i.e., losing the investment $s_t$) against one's expectations of partner reciprocity (i.e., the proportional back-transfers $r_t$). Psychologically, the investment $s_t$ can be framed as a potential loss with an immediate negative utility while the expectation of partner reciprocity denotes the predicted benefits of trust (i.e., the amount shared back by the trustee, namely, $\eta_t r_t$, where $\eta_t$ denotes the tripled amount received by the trustee at time $t$, that is, $\eta_t = 3s_t$). This expectation is represented by the expected value of the probability distribution over all possible returns $p(R_t = r_t | \mathscr{D}_{t-1})$, where $\mathscr{D}_{t-1}$ is the past trustee

**Fig. 2 | Loneliness and paranoia associations.** Both loneliness (**A**) and paranoia (**B**)––as measured by experience sampling––were significantly higher in patients with non-affective psychosis as compared to controls. **C** Overall, we observed a positive correlation between paranoia and loneliness with higher levels of paranoid thoughts in lonelier individuals. $N = 54$ participants. *significant difference; bars represent standard errors.

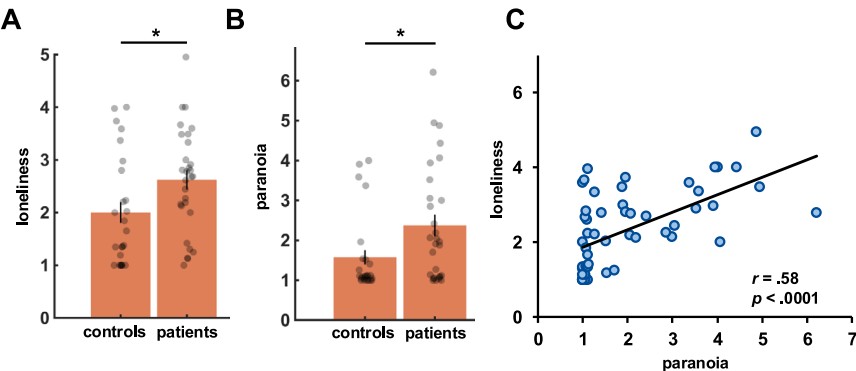

reciprocity. The investor's utility function is thus:

$$u_t(s_t, \mathscr{D}_{t-1}) = e_t - (1-\tau)s_t + \tau\left(\eta_t \int_0^1 p_t(r_t|\mathscr{D}_{t-1})r_t dr\right),$$

where $\tau = [0,1]$ is the subject-specific parameter that represents an individual's willingness to make themselves vulnerable to another based on their expectations of the other's behavior[39,54].

Moreover, to capture the sequential effects we observed in the previous behavioral analyses, we allowed for two different biases during learning. Participants might update their expectations of the partner's reciprocity differently depending on the observed reciprocity (M1; see Tab. S1), similarly to negativity/positivity biases in social learning[55–57]. Alternatively, participants might update their expectations in a non-biased manner but weight them differently depending on recent changes in partner reciprocity (M2). This parametrization captures change point effects reflecting situations in which individuals are motivated to rely on the goodwill of a partner after a single positive behavioral signal, despite the partner's previous poor reliability or, conversely, to disregard the partner's previous good reputation after a single negative behavioral signal[58–60]. Finally, we also fit expectation-only models with decay parameters added to the learning rule to account for additional recency effects in belief updating (M6-7).

As a comparison, we further fit a second class of inequity-aversion models[52], which measure how much individuals care about their own material payoff relative to that of others, and allow different weighting for advantageous and disadvantageous inequality. Let $x_t^n$ be the payoff of person $n$ at time $t$ and $x_t^m$ be the payoff of person $m$ at time $t$. We can write the utility $u_t^n$ of person $n$ at time $t$ as:

$$u_t^n(i_t, r_t) = x_t^n - \zeta \max\{x_t^n - x_t^m, 0\} - \xi \max\{x_t^m - x_t^n, 0\},$$

where $\zeta$ and $\xi$ are subject-specific parameters quantifying how much disutility an individual derives from advantageous (guilt) and disadvantageous (envy) inequality, respectively (M3). Given that payoffs are contingent on joint actions, participants can form expectations of those payoffs on a trial-by-trial basis (M4) and with potentially biased updating (M5).

Given that participants' expectations of partner reciprocity were about the possible proportional returns made by the partner, which naturally range between 0 and 1 (partner's reciprocity, $r_t = [0,1]$), participants' beliefs were modeled with a beta distribution that was updated trial by trial based on the observed returns using Bayes' rule. For a beta distribution $Beta(r_{t+1}; \alpha_{t+1}, \beta_{t+1})$ governed by the $\alpha$ and $\beta$ parameters, the posterior is updated in close-form as: $\alpha_{t+1} = \alpha_t + r_t$ and $\beta_{t+1} = \beta_t + (1-r_t)$. Finally, participants were taken to make their utility-based decisions stochastically, following a *softmax* rule:

$$p_t(s_t|\mathscr{D}_{t-1}) \propto e^{\chi u_t(s_t, \mathscr{D}_{t-1})},$$

where $\chi$ is a subject-specific parameter that captures participants' consistency with their choice utilities (in other words, decision noise).

We used Approximate Bayesian Computation (ABC) to hierarchically fit our computational model employing population Monte Carlo sampling. In ABC, we first sample candidate parameter values for group-level prior distributions from which subject-specific parameters were sampled. For the winning model, parameters for prior expectations and $\tau$ parameters (which were defined to be $0 \leq \tau \leq 1$) were sampled from independent beta distributions, while a truncated normal distribution was used for estimations of the inverse temperature ($0 \leq \chi \leq \infty$). We then use this candidate parameters to generate 100 simulations of participants' choices in the game. These simulations are then compared to the observed data using root-mean-squared deviation as a distance function (negative log-likelihood). If the distance measure was smaller than a pre-set error threshold $\varepsilon_t$, we accepted the candidate sample; otherwise we rejected it. We used a set of exponentially decreasing error thresholds $\varepsilon_t$ with values $e^{(1.5:-.1:1.5)}$. We run 100 permutations with 1000 particles each.

Model fit of the different computational models was compared using the Bayesian Information Criterion (BIC), which allows to balance model performance against model complexity. Simulation-based empirical BIC values revealed that this measure was well calibrated to identify the winning model (Fig. S2). Parameter recovery shows very good recoverability for the $\tau$ parameters with average correlations between estimated and recovered values of $\rho = .79$ for $\tau_p$ and of $\rho = .81$ for $\tau_n$. Average correlation values for the prior ($\rho = .62$) and the inverse temperature ($\rho = .64$) were slightly worse but still highly recoverable (all $ps < .00003$).

## Results

### Relationships between loneliness and paranoia

First, we observed that patients were lonelier ($M = 2.62$, $SD = 0.98$; $t_{(52)} = 2.25$, $p = 0.0287$, Cohen's $d = 0.62$, confidence intervals (CIs) = [0.07, 1.18]; Fig. 2A) and had more paranoid thoughts ($M = 2.34$, $SD = 1.46$; $t_{(52)} = 2.33$, $p = 0.024$, Cohen's $d = 0.63$, CIs = [0.011, 1.49]; Fig. 2B) than controls (loneliness: $M = 2.0$, $SD = 1.05$; paranoia: $M = 1.58$, $SD = 0.98$). Higher levels of loneliness were further associated with higher levels of paranoia across the whole sample ($r_{52} = 0.58$, $p < 0.0001$; Fig. 2C), with a descriptively (but not statistically: $Z = 1.4$, $p = 0.490$) stronger relationship in controls ($r_{23} = 0.71$, $p < 0.0001$) than patients ($r_{27} = 0.46$, $p = 0.0126$).

### Lower trust in highly-reciprocating others

We first analyzed participants' trusting behavior and their partners' reciprocity in the game. We observed that participants shared more with reciprocating partners ($\beta = 8.67$, 90% CI = [6.28, 11.18], $pd = 1$) and their trust was overall higher if their partner had higher levels of reciprocity in previous trials ($\beta = 5.11$, 90% CI = [1.69, 8.66], $pd = 0.99$). In particular, lonelier and more paranoid participants trusted highly-reciprocating partners less ($\beta = -1.25$, 90% CI = [−2.60, −0.10], $pd = 0.97$; Fig. 3A).

We also checked that trustees behaved as expected and in a similar fashion across groups and conditions, so as to rule out the possibility that potential biases in participants' behaviors were induced by our algorithm's different responses. Indeed, we observed that our pre-programmed partners increased their reciprocity in response to increases of participants' trust

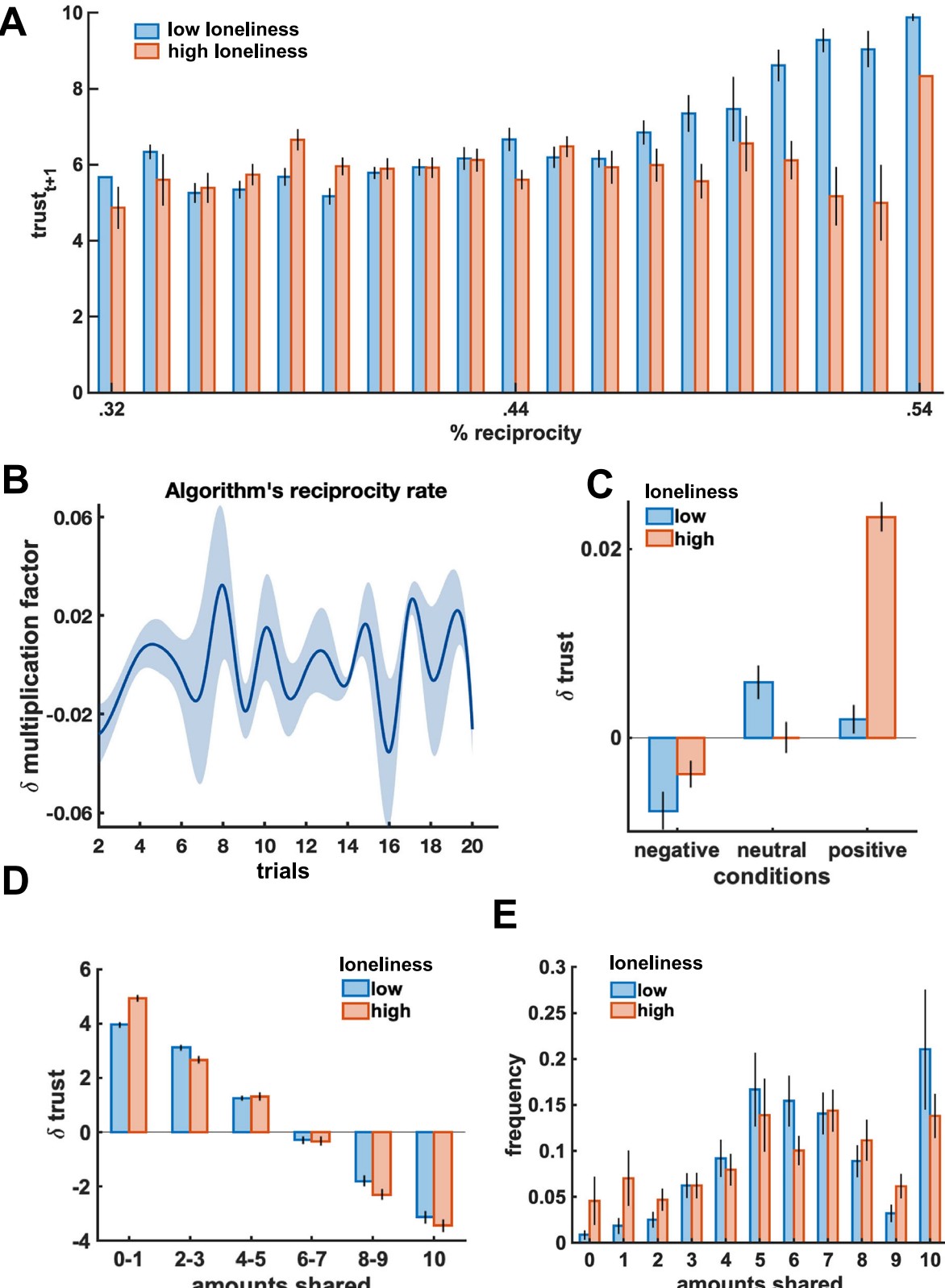

**Fig. 3 | Relationships between loneliness and trust. A** Lonelier individuals showed higher levels of trust in partners with low levels of reciprocity, while their trust was lower for highly-reciprocating partners as compared to less lonely individuals. **B** Changes in the implemented reciprocity rates (as indicated by the trialwise selected multiplication factor) show fluctuations in positive and negative reciprocity experienced by the participants. **C** Lonelier individuals showed lower levels of trust changes in the neutral and negative experimental conditions. **D** Lonelier individuals were more likely to decrease trust for higher trust levels. **E** Lonelier individuals manifested more extreme behavioral patterns, as they were more likely to share either higher or lower amounts. Note, we binned here data (e.g., high vs. low loneliness) only for visualization purposes. Bars represent standard errors. $N = 54$ participants.

($\beta = 7.36$, 90% CI = [4.20, 10.43], $pd = 1$), and they did so in similar ways both across conditions ($\beta = -0.01$, 90% CI = [$-0.16$, 0.13], $pd = 0.53$) and groups ($\beta = -0.06$, 90% CI = [$-0.48$, 0.39], $pd = 0.60$). Moreover, on average, lonelier individuals did not experience lower levels of reciprocity ($r_{52} = 0.10$, $p = 0.495$) or earnings ($r_{52} = 0.13$, $p = 0.353$) from their partners. Importantly, even though we programmed our participants' partners to be overall benevolent, their behavior constantly fluctuated over trials, leading to cases of negative reciprocity across the game (Fig. 3B).

## Trust changes over time

We then analyzed sequential and conditional effects to investigate the effects of positive and negative reciprocity (as the difference in partner reciprocity between the last two trials) on participants' trust changes (as the difference in investments between the current and the previous trial). We observed that previous lower levels of trust were more likely to predict subsequent increases of trust in the positive condition ($\beta = -0.05$, 90% CI = [$-0.10$, $-0.01$], $pd = 0.97$), suggesting that participants were slowly increasing their trust over trials and were more likely to do so when they thought their partner was benevolent. Moreover, healthy controls were more likely to increase their trust in the positive condition than patients ($\beta = -0.45$, 90% CI = [$-0.87$, $-0.04$], $pd = 0.97$).

We then turned to investigate whether this sensitivity to partner reciprocity changed over the course of the experiment. Given that the partners were actually preprogrammed to behave in the same benevolent fashion, participants should have been able to learn the true behavior of their partners over the course of the repeated interactions, accurately update their expectations, and correctly adjust their trust behavior, despite the expectation manipulation. We observed that that was indeed the case, with participants' sensitivity to partner reciprocity decreasing over time except for the negative condition where increases in partner reciprocity led to stronger increases in trust ($\beta = -1.31$, 90% CI = [$-2.52$, $-0.16$], $pd = 0.97$). We found a rather short-distance trial dependency of participants' trust changes. Neither average reciprocity change of the last three trials ($\beta = -0.02$, 90% CI = [$-0.28$, 0.25], $pd = 0.55$) nor a change in reciprocity three trials back ($\beta = 0.11$, 90% CI = [$-0.13$, 0.36], $pd = 0.76$) significantly predicted the most recent change of trust, suggesting recent history dependencies of change points, in line with previous evidence on recency effects of partner reciprocity on trusting behaviors[58,61].

## Lonelier and more paranoid individuals are more sensitive to partner reciprocity

We then asked whether differences in participants' responsivity to partner reciprocity were further modulated by individual's levels of loneliness. We observed that greater feelings of loneliness were associated with lower sensitivity to partner reciprocity, particularly in the neutral condition ($\beta = -0.34$, 90% CI = [$-0.54$, $-0.14$], $pd = 1$), suggesting that lonelier individuals changed their trust in that condition less than less lonely individuals (Fig. 3C). Importantly, lonely individuals' changes of trust were contingent on their previous trust levels (Fig. 3D). In particular, lonely individuals were more likely to decrease their trust had they manifested higher levels of trust in the previous trial, especially in the negative condition ($\beta = 0.05$, 90% CI = [0.01, 0.08], $pd = 0.99$) and despite having experienced positive reciprocity from the partner ($\beta = 0.47$, 90% CI = [0.02, 0.90], $pd = 0.96$). This suggests that while less lonely individuals might share little out of cautiousness but are ready to increase their trust upon signs of a partner's goodwill, lonelier individuals might be more suspicious, and their lower investments might more directly signal distrust.

Stronger reductions in trust observed in lonelier individuals could be explained by at least two mechanisms: a punitive mechanism by which lonely individuals might have wanted to punish their partner for being uncooperative and unkind; or a protective mechanism by which lonely individuals might have feared a betrayal and hence pre-emptively reduced their vulnerability to the partner. A punitive mechanism might be triggered if a trustee responds with an inadequate level of reciprocity to a participant's sign of cooperation (i.e., an increase of trust). As a consequence, a decrease of

trust in response to positive reciprocity could signal punitive behavior. However, as shown above, despite cases of breaches of trust, the trustees were on average reciprocating participants' increases of trust with positive reciprocity and overall, participants' trust changes positively correlated with changes in partner reciprocity ($\rho_{224} = 0.18$, $p = 0.0083$), suggesting that participants were seeing their trust honored by their partners and were responding by increasing their own trust.

On the contrary, a vulnerability mechanism could be triggered by the fear of experiencing breaches of trust (i.e., negative reciprocity). This fear should be greater in those interactions where there was more at stake, such as interactions with highly reciprocating partners whose negative reciprocity meant greater losses. Indeed, interactions with highly-reciprocating partners involved higher levels of trust ($\rho_{224} = 0.32$, $p < 0.00001$) and since we pre-programmed all partners not to share more than half of what they received (a fairness upper bound), highly-reciprocating partners in our experiment were more likely to show negative reciprocity in future trials ($\rho_{224} = -0.80$, $p < 0.00001$). This, together with lonely individuals' lower trust levels and weaker sensitivity to positive reciprocity in interactions with highly-reciprocating trustees, suggests that lonely individuals' trust was more strongly modulated by a fear of being vulnerable to a partner's betrayal. This could also explain why lonelier individuals showed patterns of trusting behaviors that more closely resembled an "all-or-nothing" strategy (Fig. 3E). They might have shared high amounts to manifest strong cooperation but, as the partner increased their level of reciprocity (Fig. 3A), they might have feared a betrayal and either completely withdrew from the interaction (i.e., sharing nothing) or reduced their trust to the lowest levels (i.e., sharing the smallest amounts) as to probe the partner's "true" reciprocity (Fig. 3E).

Such a state of greater vulnerability might have been due to higher suspiciousness about a partner's reciprocity. We hence tested whether paranoid thinking moderated these patterns of trusting behavior. We found that especially in the negative condition, individuals with lower levels of paranoia were more likely to increase their trust in response to a partner's positive reciprocity had they manifested lower levels of trust in the previous trial ($\beta = 0.03$, 90% CI = [0.01, 0.06], $pd = 0.98$), indicating that suspiciousness might have played a role in why participants shared low investment and were less responsive to positive reciprocity. However, we did not find any significant interaction between loneliness and paranoia in predicting participants' trust changes ($\beta = -0.05$, 90% CI = [$-0.19$, 0.11], $pd = 0.70$).

## Expectations of reciprocity and willingness to trust

Differences in trusting behaviors could be explained by at least two different mechanisms: participants might be biased in updating their expectations of the partner's reciprocity, or they might accurately learn the partner's reciprocity but differently weight those expectations. To disentangle these routes, and to understand the inner working of how partner's behavior shapes participants' choices, we developed a formal model operationalizing how participants formed expectations of their partners and weighted those expectations into their decisions. We also estimated participants' initial expectations of partner reciprocity, to investigate biases in those initial expectations and how they were updated based on the partner's reciprocal behavior observed later on in the experiment. Model comparison (Fig. S1 & Table S2) revealed that the winning model was model M2 with two $\tau$ parameters weighting participants' expectations differently after positive and negative reciprocity (Fig. 4A), suggesting a weighting bias rather than a learning bias.

Analyses of model parameters revealed that the estimated prior expectations laid within the fairness range with an average expectation of partner reciprocity of $M = .44$ ($SD = .10$) and an 90% credible interval between 0.30 and 0.61. This suggests that participants expected their partner to share back a fair proportion of what they received. Moreover, average prior expectations estimated by the model strongly correlated with participants' trust in the first trial ($r_{52} = 0.83$, $p < 0.00001$; Fig. 4B), suggesting that the model well captured the initial, individual expectations of partner

**Fig. 4 | Model-based expectations and willingness to trust. A** Graphic representation of the vulnerability model. **B** Prior expectations strongly predicted participants' first trust decisions. **C** Participants tracked their partners' reciprocity well as shown by model-based beliefs ($\rho$) about partners' reciprocity and observed reciprocal behavior. **D** The average of participants' $\tau$ parameters strongly correlated with their average trust (i.e., investments) in the game. **E** Probability change for the previously chosen action after negative ($\tau_n$) and positive ($\tau_p$) in reciprocity in the vulnerability model. **F** Same probability changes as a function of high and low levels of loneliness. $N = 54$ participants. *significant difference.

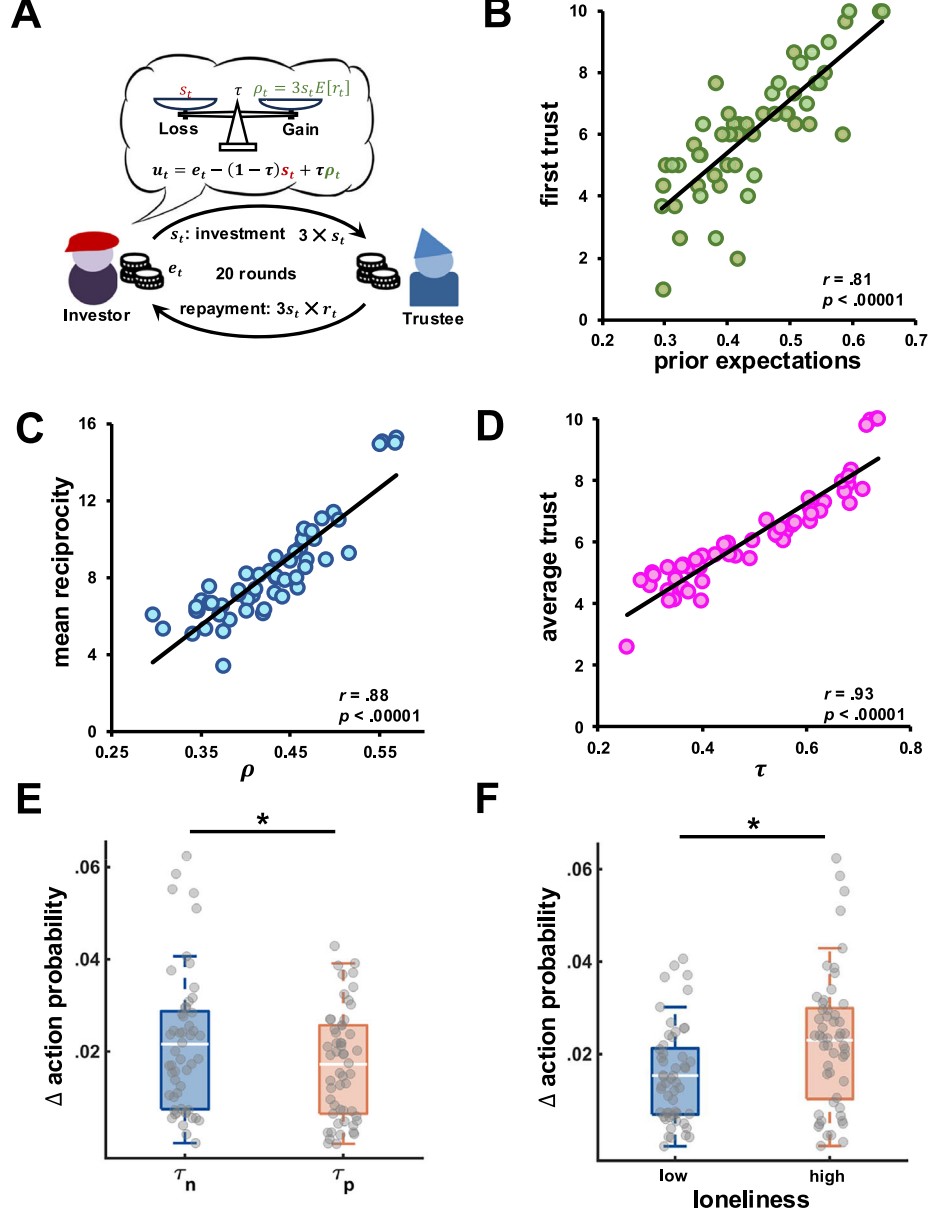

reciprocity. Finally, we observed an interaction between group and condition, with patients having more negative prior expectations in the neutral condition ($\beta = -0.09$, 90% CI = $[-0.16, -0.01]$, $pd = 0.97$) as compared to controls. No significant interaction was found between group and the positive condition ($\beta = -0.05$, 90% CI = $[-0.12, 0.02]$, $pd = 0.87$).

Moreover, model-based parameter values suggest that patients weighted expectations of partner reciprocity more than controls after both positive ($\tau_p$: $\beta = 0.10$, 90% CI = $[0.01, 0.19]$, $pd = 0.97$) and negative reciprocity ($\tau_n$: $\beta = 0.10$, 90% CI = $[0.01, 0.19]$, $pd = 0.96$). Moreover, after positive reciprocity, participants weighted expectations of partner reciprocity significantly more in the neutral ($\beta = 0.06$, 90% CI = $[0.002, 0.12]$, $pd = 0.96$) than negative condition but not in the positive condition ($\beta = 0.03$, 90% CI = $[-0.03, 0.09]$, $pd = 0.76$). No condition or group differences were observed for expectation weighting after negative reciprocity ($\tau_n$). Finally, participants' expectations of partner reciprocity as estimated by the model strongly correlated with the actual reciprocity observed ($r_{52} = 0.88$, $p < 0.00001$; Fig. 4C) and average $\tau$ parameters strongly correlated with participants' trust in the experiment ($r_{52} = 0.93$, $p < 0.00001$; Fig. 4D), suggesting that our operationalization of an individual's willingness to trust strongly relate to their actual trusting behavior in the game.

Further, an analysis of the policy profiles estimated by the vulnerability model shows that participants were changing their policies (i.e., probability distribution over investments) differently after positive and negative reciprocity (Fig. 4E). Specifically, the action probability (i.e., the utility-based probability of choosing a given investment) for the investment chosen by the participant at time $t - 1$ more strongly changed at time $t$ after observing a decrease in reciprocity ($F_{(1,103)} = 12.80$, $p = .0005$, partial $\eta^2 = .84$, CIs = $[-0.016, -0.002]$). These policy changes were greater for lonelier ($F_{(1,103)} = 6.25$, $p = .0139$, partial $\eta^2 = 0.06$, CIs = $[-0.009, -0.003]$; Fig. 4F) and more paranoid individuals ($F_{(1,103)} = 5.98$, $p = .016$, partial $\eta^2 = 0.06$, CIs = $[-0.011, -0.003]$), corroborating the above model-agnostic findings that lonelier individuals manifested more extreme behavioral patterns (i.e., were more likely to share either everything or nothing).

## Lower willingness to trust in lonelier and more paranoid individuals

We then investigated whether the degree to which individuals weighted their expectations of partner reciprocity was modulated by their levels of loneliness and paranoia. We found that lonelier individuals weighted their expectations of the partner's behavior in response to changes in partner

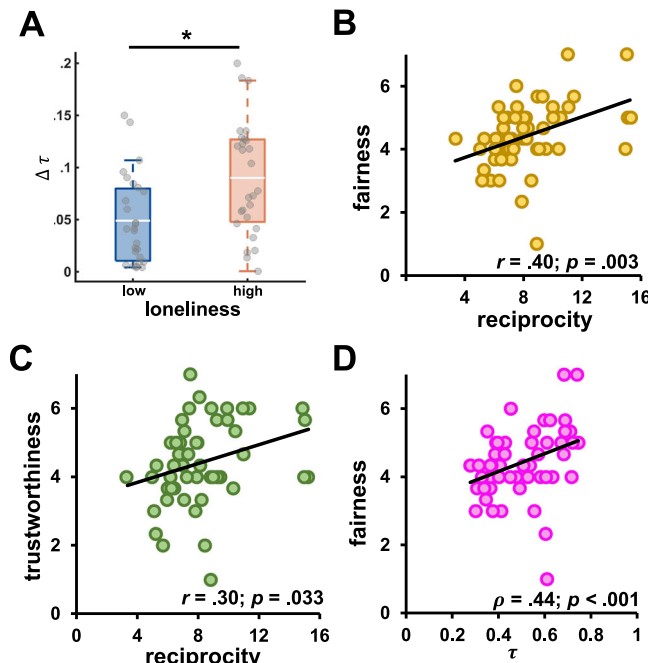

**Fig. 5 | Expectation weighting in lonely individuals and post-game impressions.**
**A** The difference in willingness to trust after positive and negative reciprocity (i.e., $\Delta\tau$) was greater in lonelier participants (N = 54 participants). Participants' overall impressions of partners' fairness (**B**) and trustworthiness (**C**) correlated with observed reciprocal behavior in the investment game (N = 53 participants).
**D** Average willingness to trust in the investment game as estimated by the vulnerability model ($\tau$) was positively associated with more positive fairness impressions of a partner (N = 53 participants). * significant difference.

reciprocity differently than less lonely individuals ($t_{(52)} = 3.08, p = .003$, Cohen's $d = 0.84$, CIs = [0.014, 0.068]; Fig. 5A). This different expectation weighting led lonelier individuals to show higher variability in their responses to changes in partner reciprocity ($t_{(52)} = 2.41, p = .019$, Cohen's $d = 0.66$, CIs = [0.013, 0.135]). Moreover, lonelier individuals had an overall greater willingness to trust after both positive ($\tau_p: \beta = 0.07$, 90% CI = [0.01, 0.13], $pd = 0.96$) and negative reciprocity ($\tau_n: \beta = 0.08$, 90% CI = [0.02, 0.15], $pd = 0.98$). On the contrary, more paranoid individuals were more willing to trust after positive ($\tau_p: \beta = 0.10$, 90% CI = [0.02, 0.15], $pd = 0.97$) but not after negative reciprocity ($\tau_n: \beta = 0.08$, 90% CI = [−0.01, 0.17], $pd = 0.94$). However, lonelier individuals with higher levels of paranoia manifested a lower willingness to trust after positive reciprocity ($\tau_p: \beta = −0.04$, 90% CI = [−0.06, −0.002], $pd = 0.96$).

### Expectations of reciprocity and willingness to trust contribute to more negative impressions

Participants' ratings after the investment game were consistent with our expectation manipulation. Specifically, partners were reported to behave more positively in the neutral (trustworthiness: $\beta = 0.88$, 90% CI = [0.23, 1.53], $pd = .99$; fairness: $\beta = 0.84$, 90% CI = [0.11, 1.57], $pd = 0.97$; generosity: $\beta = 0.75$, 90% CI = [0.02, 1.46], $pd = .96$) and positive (trustworthiness: $\beta = 1.24$, 90% CI = [0.60, 1.89], $pd = 1$; fairness: $\beta = 1.04$, 90% CI = [0.30, 1.76], $pd = 0.99$; generosity: $\beta = 0.79$, 90% CI = [0.08, 1.51], $pd = 0.96$) conditions than in the negative condition, showing a linear increase in the positivity of partner impressions. A significant group effect shows that patients rated their partners as more generous ($\beta = 1.39$, 90% CI = [0.52, 2.30], $pd = 0.99$) than controls, while no significant group differences were found for fairness ($\beta = 0.75$, 90% CI = [−0.07, 1.58], $pd = 0.93$) and trustworthiness ratings ($\beta = 0.70$, 90% CI = [−0.18, 1.58], $pd = 0.91$). Moreover, fairness impressions were significantly more negative in individuals with higher levels of paranoid thinking ($\beta = −0.92$, 90% CI = [−1.55, −0.25], $pd = 0.99$) and a significant interaction shows that this effect was positively

modulated by individuals' levels of loneliness ($\beta = 0.27$, 90% CI = [0.07, 0.48], $pd = 0.98$), suggesting that loneliness and paranoia synergistically modulate social expectations and behaviors.

The formation of more negative impressions of a partner might have depended on both more negative initial expectations and a lower willingness to trust, which could have led to a less successful interaction. As a consequence, partners might have reciprocated less with our participants, inducing more negative impressions. Indeed, participants' impressions positively correlated with average partner reciprocity (fairness: $r_{51} = .40$, $p = 0.003$; trustworthiness: $r_{51} = .30$, $p = 0.033$; generosity: $r_{51} = 0.40$, $p = 0.004$; Fig. 5B, C). Corroborating this interpretation, we observed that more positive fairness impressions were associated with more positive prior expectations of partner reciprocity ($\rho_{51} = 0.33$, $p = 0.017$) and a greater willingness to trust after both positive ($\rho_{51} = 0.45$, $p < 0.001$) and negative reciprocity ($\rho_{51} = 0.39$, $p < 0.004$; Fig. 5D). Overall, these findings suggest that the formation of more negative impressions depended on both unfavorable initial expectations before the social interaction and a reduced willingness to trust a partner during the social interaction.

### Discussion
In this study, we investigated the relationship of loneliness and paranoia with an individual's willingness to trust and their expectations of others' reciprocity in a sample of individuals with and without a psychotic disorder. We did that by manipulating participants' initial expectations of their three partners (trustees) in an iterated investment game, and investigating their behavior change based on the reciprocity of preprogrammed benevolent partners. We found that the expectation manipulation successfully biased participants' willingness to trust and their impressions of their partners (see also ref. 30). Computational modeling revealed that even though lonelier individuals were generally more willing to trust their partners, they manifested lower levels of trust in highly-reciprocating partners, and their willingness to trust decreased more as a function of their paranoid thinking. Importantly, these effects were observed in both patients and controls, suggesting loneliness and paranoia negatively modulated social expectations and behaviors in all participants.

Patients with psychosis have previously been observed to be six times more likely to experience loneliness and social exclusion[31,62,63], and previous work has suggested that social isolation in individuals with paranoid thoughts worsens paranoid symptoms by fostering retention of unusual ideas[64]. In our study, we observed a strong relationship between loneliness and paranoia in individuals with and without non-affective psychosis. This finding is in line with the idea that paranoia should be understood as a continuum from non-clinical to clinical states, and that feelings of loneliness might induce a spiral of negative thoughts and maladaptive cognitions that could prelude and foster clinical symptoms[29,45,64,65]. Further, the difference in the strength of the association between loneliness and paranoia could reflect differences in how loneliness and paranoia are influenced by situational factors in the clinical and general population[59].

Similarly, the relationship between loneliness and paranoid thoughts in the control group suggests that these two experiences may co-occur even in the absence of clinical psychopathology. This relationship and its effects on social behaviors fundamental for relationship building––such as trust––highlight shared psychological and cognitive patterns that underlie loneliness and paranoia in both patients and the general population. These findings underscore the importance of addressing maladaptive social-cognitive processes transdiagnostically, especially in preventive contexts. While the current findings cannot directly speak to the directionality of the relationship, a recent temporal analysis between loneliness and paranoia supports the hypothesis that reducing loneliness decreases paranoid thinking over time[29], but not vice versa. Similarly, preliminary experimental evidence also suggests that experimentally decreasing loneliness leads to a reduction of paranoia[28]. Further studies are needed to investigate the causal and temporal mechanisms underlying the loneliness-paranoia association.

Lonelier individuals were more likely to manifest high levels of trust than less lonely individuals. This is consistent with previous work on one-

shot trusting interactions showing that despite more negative trustworthiness expectations, lonelier individuals trusted their partners more than less lonely individuals[42]. However, our study also revealed that lonelier individuals were more sensitive to changes in partner reciprocity, in line with the idea that they manifest a heightened alertness to social partners[66]. In particular, decreases of partner reciprocity led lonelier individuals to reduce their trust more drastically than less lonely individuals. Interestingly, these patterns were particularly pronounced in the negative condition and when partners had high reciprocation rates, so that on average, lonelier individuals trusted highly-reciprocating partners less. Sequential analyses demonstrated that as highly-reciprocating partners in our experiment were more likely to show smaller increases in reciprocity in the future, participants interacting with them were more likely to experience negative reciprocity. In other words, those partners were more likely to be perceived as reducing their cooperation and thus breaching participants' trust. The more drastic reduction of trust observed in lonelier individuals in response to these breaches of trust by highly-reciprocating partners might have induced a spiral of distrust undermining the potential of a successful trusting interaction. These effects could be explained by different mechanisms.

On the one hand, drastic reductions of trust after negative reciprocity could be seen as a form of retaliation or punishment by the investor who wants to signal their disappointment with the observed level of reciprocity and stimulate higher future returns. Indeed, refusing to share money in the investment game––classically interpreted as a self-interested choice––can also signal irritation and disrespect, and hence be used as a form of social punishment[58,67]. In line with this, we observed that participants with higher levels of paranoid thinking were more likely to reduce their trust after observing a partner's negative reciprocity. This suggests that individuals who perceived a decrease in reciprocity by their partner as a more serious breach of their trust reacted with greater distrust. This aligns well with previous evidence that paranoid participants are more willing to punish less generous partners and that their punishment was mediated by a tendency to attribute harmful intentions to others[33].

On the other, drastic reductions of trust after negative reciprocity could have been the result of an attempt to reduce one's vulnerability to the partner. Indeed, we found that lonelier individuals had overall higher levels of trust than less lonely individuals, but only in partners with lower levels of reciprocity. Since participants stood to lose more to highly-reciprocating partners, they might have felt more vulnerable when interacting with them. This is corroborated by evidence that highly-reciprocating partners were more likely to manifest negative reciprocity (i.e., decreases in reciprocity). As a result, lonelier individuals might have maintained lower levels of trust to reduce their exposure to a betrayal (i.e., negative reciprocity). This interpretation is in line with previous evidence that lonelier individuals manifest greater rejection sensitivity[68] and stronger sensitivity to negative than positive feedback about others' social behaviors[44]. Importantly, these behavioral patterns also point to the fact that lonelier individuals might be biased in representing their partners' intentional states. For instance, given lonelier individuals' overall negative expectations of other people's behaviors, highly-reciprocating trustees' cooperation must have been hard to believe (in Bayesian terms: had low posterior probability) and was likely interpreted as indicative of strategic thinking and malicious intent rather than genuine kindness.

Computational modeling revealed that lower levels of trust were not due to inaccurate or biased learning patterns but to lonelier individuals' lower willingness to rely on their expectations of reciprocity from their partners that they formed during the course of the interaction. In particular, participants' willingness to trust changed based on the most recent changes in partner reciprocity and lonelier individuals manifested a bigger difference in their willingness to trust after changes in partner reciprocity. This difference induced greater changes in action probabilities, which contributed to a higher variability in lonelier individuals' trusting behavior. These action probability changes indicate a greater variability in lonelier individuals' behavioral strategy that, despite

unbiased expectations, was responsible for their abrupt behavior changes and heightened sensitivity to changes in partner reciprocity. This is consistent with model-agnostic analyses showing a prominent "all-or-nothing" strategy, with lonelier individuals more likely to either share all of their endowment or none. Such a strategy necessarily leads lonely individuals to manifest a greater behavioral volatility that could be interpreted as unreliable behavior by their social partners, thereby inducing more negative trait impressions in others (e.g., of being an untrustworthy person) that undermine their attempts to establish successful social relationships[56,69].

Further, suspiciousness about the partner's true reciprocity could have reduced lonely individuals' willingness to trust. Indeed, we observed a lower willingness to trust in lonelier individuals who also reported higher levels of paranoia after positive reciprocity. This suggests that lonelier individuals with higher levels of paranoid thinking were not taking the chance to build on signals of cooperation to establish a more successful trusting relationship[58]. Together, these results suggest that the negative relationship of loneliness and paranoid thinking with social behaviors and expectations is due to a synergistic feedback mechanism based on biased attributions of intent. In particular, lonely individuals might initially be prone to interact and connect with others (as shown by their overall greater willingness to trust and higher likelihood to share high amounts) but their suspiciousness of others and possible paranoid interpretations of partner reciprocity might induce malevolent intent attributions (such as excessive attributions of harm intent)[70,71] or probing behaviors (such as reductions of trust to its lowest levels) that foster distrust and social withdrawal[72]. Specifically, lower willingness to trust in lonelier individuals with higher levels of paranoia could lead to overall less cooperative and trusting interactions for three reasons. First, an inappropriate increase of trust might signal a suspicious cautiousness that could discourage a cooperative partner. Second, a lack of attempts to persuade a less cooperative partner might represent a missed opportunity to provide needed evidence of one's trustworthiness. Third, probing behaviors, especially in combination with behavioral unreliability, might reveal both distrust and uncertainty that could irritate a benevolent partner, scare a cautious one, and alert the strategic.

These dynamics are supported by participants' overall impressions of their partners' fairness and trustworthiness after the investment game. Notably, more negative impressions of a partner's fairness were associated with lower levels of reciprocity. These reduced reciprocity levels, in turn, were linked to both more negative prior expectations of the partner's behavior and a decreased willingness to trust following changes in the partner's reciprocity. These findings align with previous evidence that paranoid individuals playing against unfair partners attribute more harmful intent to them[73] and were able to escape a spiral of negative relationships only when interacting with consistently-fair partners[74]. In trusting interactions, a partner's reciprocity is inherently dependent on an individual's initial willingness to trust. Therefore, high levels of reciprocity can only emerge when one is prepared to extend significant trust. A diminished readiness to trust may thus set in motion a cycle that contributes to increasingly negative social experience.

## Limitations

Future studies are needed to further explore the complex dynamics between expectations and behaviors in lonely individuals and overcome some of the limitations of this study. For instance, we employed a simple algorithm to generate our trustees' behaviors, which allowed us to only investigate simple sequential effects of positive and negative reciprocity. More sophisticated algorithms to generate richer behaviors of social partners in economic games, for example, by using generative modeling approaches, might help shed light on interindividual differences in biases in social inferences and behaviors, as well as their relationships with loneliness and paranoia. In addition, our study had a relatively low sensitivity (in relation to more subtle effects revealed in the literature) and future study would need to replicate our results with a bigger sample size, potentially also using online populations (e.g., individuals with subclinical levels of paranoia), given the

transdiagnostic nature of the effects of loneliness and paranoia on social behaviors observed in this study. Moreover, future studies would need to test degree to which state- and trait-like measures of loneliness and paranoia capture different dynamics in lonely individuals' social behaviors. For example, given the heightened sensitivity to rejection in lonely individuals, some forms of social punishment (e.g., distrust but also second-party punishment) could be exacerbated in periods of high levels of momentary loneliness. Finally, longitudinal studies, combined with computational modeling, could provide insights into how the mechanistic processes highlighted in this study change over time and predict both a severe development of lonely states and other mental health issues such as paranoid-like reasoning patterns.

## Conclusions

Taken together, in this study, we showed that lonelier individuals were more willing to trust their partners but were also more sensitive to changes in partner reciprocity, showed more variable trusting behavior, and were less likely to rely on their expectations of a partner's reciprocity, further suggesting that loneliness and suspiciousness jointly modulate a person's social expectations and behaviors. These trusting patterns might have contributed to a self-reinforcing mechanism that led to overall more negative impressions and reduced trust, which fed forward into less cooperative and successful social interactions. These results highlight the difficulties for lonely individuals to establish successful trusting relationships, even when the partner is benevolent. Importantly, these effects were observed in both patients and controls, suggesting that loneliness and paranoia operate at subclinical levels of suspiciousness via psychological mechanisms that hamper a person's attempts to establish a supporting and nurturing social network.

## Data availability

The behavioral data and computational modeling results are openly available on OSF.

## Code availability

The code is openly available on OSF.

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

## Acknowledgements

The authors thank the participants of the DECOP project and the National Institute for Health Research (NIHR) Clinical Research Organizations (CROs), as well as Farah Yakub, Marieke Helmich, Marina Volguraki, Katie Wong, Mathew Harvey, and Tracy Bobin for their support with testing and recruitment.

## Author contributions

G.B.: Conceptualization; software; methodology; formal analysis; visualization; writing – original draft; writing – review and editing. M.K.: Methodology; visualization; writing – review and editing. E.H.: Investigation; data curation; writing – review and editing. A.-K.F.: Conceptualization; data curation; funding acquisition; resources; supervision; project administration; writing – review and editing.

## Competing interests

The authors declare no competing interests.
