## [Transparent Peer Review file · Communications Psychology]

Willingness to trust is reduced by loneliness and paranoia

Corresponding Author: Dr Gabriele Bellucci

Version 0:

Decision Letter:

Dear Dr Bellucci,

Thank you for your patience during the peer-review process. Your manuscript titled "Willingness to trust is reduced by loneliness and paranoia" has now been seen by 2 reviewers, whose comments are appended below. You will see that they find your work of some potential interest. However, they have raised quite substantial concerns that must be addressed. In light of these comments, we cannot accept the manuscript for publication, but would be interested in considering a revised version that fully addresses these serious concerns.

We hope you will find the Reviewers' comments useful as you decide how to proceed. Should additional work allow you to address these criticisms, we would be happy to look at a substantially revised manuscript. If you choose to take up this option, please highlight all changes in the manuscript text file, and provide a detailed point-by-point reply to the reviewers.

Editorially, we consider it crucial that the experimental design and analyses are appropriate for the testing of punishment vs fear of vulnerability hypotheses in the revised manuscript. Please provide evidence that the measures used for paranoia and loneliness were standardized validated measures; if this wasn't the case, we would require a replication study in a well-powered (healthy) control sample using a validated measure of paranoia as a precondition for further review. We also strictly recommend the use of the combination of the 3-item UCLA scale and the single-item measure, as suggested in [https://www.ons.gov.uk/peoplepopulationandcommunity/wellbeing/compendium/nationalmeasurementofloneliness/2018/recommendednationalindicatorsofloneliness](https://www.ons.gov.uk/peoplepopulationandcommunity/wellbeing/compendium/nationalmeasurementofloneliness/2018/recommendednationalindicatorsof Loneliness).

Equally importantly, we require evidence that the study is sufficiently powered to detect the smallest theoretically or pragmatically meaningful effect (SESOI). In other words, we require a sensitivity analysis with a justification for the SESOI (not a post-hoc power analysis based on the observed effect size in your study [cf. Lakens, 2022, <https://doi.org/10.1525/collabra.33267>]).

If you decide to submit a revised manuscript, please ensure you follow our statistical guidelines when reporting statistics (<https://www.nature.com/commspsychol/submit/submission-guidelines#statistical-guidelines>). Please note in particular our requirements for the reporting and interpretation of null-results. Non-significant findings derived from null-hypotheses significance tests should be reported in full, but may not be interpreted. Where you interpret null results, this interpretation must be based on Bayes Factors or equivalence tests.

We understand that these requests are substantive and you may instead choose to resubmit your manuscript elsewhere. If this is the case, we would be grateful if you notified the editorial office so that we can close your file.

I am attaching a checklist that details critical reporting requirements for the revised manuscript. Please attend to each item and ensure your manuscript is fully compliant. We are requesting that your manuscript aligns with these requirements as this facilitates the evaluation of your manuscript, reducing delays in re-review and potential future acceptance. If your revised manuscript is not aligned with these requests on major issues, such as those concerning statistics, it may be returned to you for further revisions without re-review. Additional information can be found in our style and formatting guide Communications Psychology formatting guide.

If the revision process takes significantly longer than five months, we will be happy to reconsider your paper at a later date, provided it still presents a significant contribution to the literature at that stage.

Please use the following link to submit your

- revised manuscript,
- point-by-point response to the referees' comments,
- cover letter (as a separate document),
- the Editorial Policy Checklist (see below),
- the Reporting Summary (see below), and
- the completed Editorial Request Table (attached):

Link Redacted

Thank you for the opportunity to review your work.

Best regards,

Troby Lui

Troby Lui, PhD
Associate Editor
Communications Psychology

REVIEWER EXPERTISE:

Reviewer #1: loneliness, paranoia/psychosis

Reviewer #2: paranoia/psychosis, trust game

REVIEWER REPORTS:

Reviewer #1 (Remarks to the Author):

The manuscript examined the behavioural patterns and responses in loneliness and paranoia underlying social uncertainty, as captured by the investment game. The sample consisted of 54 participants (29 participants with non-affective psychosis and 25 healthy controls). The use of computational modelling revealed interesting behavioural tendencies (e.g. trust) in responses to variations in payoffs and expectations of players' responses (i.e., players were told to be benevolent or malevolent, or without this piece of information), which were found to be associated with individual differences in loneliness and paranoia. The experiment (i.e., investment) was sophisticatedly designed, the inclusion of both clinical and non-clinical participants in this sample, the use of an advanced modelling approach are merits of this study. These findings suggest biased social-cognitive processing belief formation and updating may underlie the perpetuation of loneliness and paranoia. My comments are as follows:

Major comment:

The manuscript would benefit from the improvement of theoretical clarity guiding this investigation. The bidirectional relationship between loneliness and paranoia is theorized and expected, while more studies, including experimental studies using loneliness manipulation and longitudinal studies, support loneliness as an antecedent of paranoia. The Introduction mentions several biased social-cognitive processes, lack of social motivation, lower trust and negative social expectation, but how these processes influence or are influenced by loneliness and paranoia is not entirely clear. Some descriptions are too vague (e.g., "various mechanisms might explain these thinking patterns", 4th paragraph, Introduction), that whether these mechanisms are shared between loneliness and paranoia, or unique to loneliness or paranoia warrants further clarification. I'm also confused about how these processes were represented or operationalized in the investment game paradigm.

In particular, loneliness and paranoia are distinct but interrelated concepts, the conceptual clarity should be carefully expressed. Meanwhile, as mentioned in the 1st paragraph of the Introduction, loneliness is believed to be a transdiagnostic phenomenon across psychopathologies, such as social anhedonia, depression and social anxiety, as well as other psychotic experiences beyond paranoia. Therefore, we expect lonely individuals do not necessarily have paranoid thinking, so both concepts should not be taken parallel, from the perspective of psychopathology and individual differences. Some of the arguments highlight the research gap appear only in the Discussion, and I would expect them to appear earlier to set the stage in the Introduction for the readers' easier understanding.

Other comments:

Abstract:

- The abstract should also briefly mention the study design, including the sample and the behavioural task.

Introduction:

- The Introduction should end with brief descriptions of the experimental design, the behavioral paradigm used, key variables/ parameters to look at, and operationalization of the variables, the use of both clinical and non-clinical samples, and also studies hypotheses. At the current stage, the linkage between the Introduction and Results needs to be strengthened, as the Methods only appear at the end of the manuscript and you should expect the readers could grasp the core of the study design when they read the Results right after the Introduction.

Methods and Results:

- It's not clear how the variables for analysis were measured and matched with the variables used in the analysis (e.g. , reciprocity, trust). As there are quite many parameters (e.g., ratings of trustworthiness, the amount invested in each trial), it is not straightforward to see how these parameters were treated and matched with variables for analysis. Maybe a table listing these explaining the variables and parameters would facilitate comprehension.

- Was there any a-prior sample size calculation?

- It's interesting to see that participants were explicitly told about the interacting partners' predicted behavioural tendencies (i.e., to be benevolent or malevolent) before the trials. How would this experimental manipulation interact with the probabilistic payoffs in shaping participants' investment decisions? Cognitive models of paranoia/ persecutory delusions have posited the unique role of negative-others schemas, as well as hostility attribution bias, in the development and maintenance of paranoia. It should be expected when engaging in the experiments, participants were primed about the hostility of the players, which may further activate negative-other schemas to trigger mistrust or paranoid thinking (and indeed participants with high paranoia may be less willing to buy in the information that the players were with good intention). How would this (and its impacts on behaviors such as trust and belief formation/ updating, either intended or unintended), be modelled in the analysis?

- I suppose the authors consider loneliness and paranoia as more trait-like aspects of these phenomena in their conceptualization of research questions (e.g. individuals who are more lonely/ paranoid were hypothesized to show less trust when they expected their partners to cheat or not to reciprocate). However, well-established measures of loneliness (e.g. UCLA Loneliness Scale) and paranoia (e.g. Green et al. Paranoid Thoughts Scale) were not mentioned (and maybe not administered?) throughout the manuscript. Rather, state-like, momentary measures of loneliness and paranoia were captured with the experience sampling method. I guess loneliness and paranoia in this study were conceptualized with the mean ratings of the experience sampling measures of both phenomena respectively (please confirm!). If so, there is a disjoint between theory/ research questions and the measures of loneliness and paranoia. The inclusion of well-validated measures of loneliness and paranoia is important, as these are the main variables of interest. Please clarify this methodological issue.

Discussion:

- In the second paragraph of the Discussion, the authors suggest 'the strong relationship between loneliness and paranoia in our participants without a psychotic disorder might point to the importance of tackling loneliness to alleviate paranoid thinking and vice versa', the logic of the argument didn't read well. Please rewrite to improve coherence and logic. Indeed, the directionality of the relationship between loneliness and paranoia can't be tested in this study.

Overall:

- Please avoid using casual language such as "xxx affect xxx", "the impact of..." and "the effect of...", as causal conclusions from this design are limited.

Reviewer #2 (Remarks to the Author):

This manuscript examines the influence of loneliness and paranoia on trust behaviors as the investor in an iterated trust game in healthy controls and patients with psychosis. They additionally applied a computational vulnerability model to measure concerns of vulnerability (willingness to trust) and asymmetrical responses to a benevolent partner's reciprocity. The stated aim to provide insight into how loneliness and paranoia influence negative impression formation; however, there are several ideas presented that do not appear cohesive throughout the manuscript, although I am not sure if this is due to inconsistency in ideas or writing style. The overarching aim of the study is interesting and valuable to understanding distrust and factors that may contribute to its development, but I have several concerns. I am not convinced that the current analyses support the claims made. Below are some concerns with the framing of the current document as well as thoughts to consider for a future manuscript.

1. In order to put the methods at the end, more needs to be explained in simple terms in the introduction and results sections. It doesn't need to be all the details, but enough that a person can understand the game, the manipulation, how many participants were included, and the definitions of independent and dependent variables.

2. With that said, I don't think the vulnerability model needs to be explained in the results so thoroughly; instead define the variables enough so a reader knows what they represent, and then the math can be in the methods. Stay consistent with how you refer to them as well, as sometimes they are referred to as phrases and sometimes, they are referred to as characters. My personal preference is to use a word or phrase with the character in parentheses (i.e., willingness to trust ()). In addition, I would like more explanation for what action probabilities represent in the model.

3. The 4th paragraph of introduction is unclear to me in its current form. From what I understand, you are comparing hypotheses to explain suboptimal behavior in lonely individuals: increased punishment vs. fear of vulnerability. Conceptually that makes sense, but the wording took a while to figure out what was meant, and it felt unclear if or how these hypotheses were being directly compared.
4. I think the fear of vulnerability hypothesis fits with the end of the results & discussion section, which is that lonelier & more paranoid individuals give less initially and don't respond as strongly to positive reciprocity from the partner, but it wasn't clear how it fit in the larger context of the paper due to differences in language use. The authors argue that this behavior leads to lower reciprocity from the partner (as it's programmed to do), causing a feedback loop. For what it's worth, I think this paragraph is stronger. In particular, showing that prior expectations of reciprocity was associated with fairness impressions and willingness to trust in the game suggests that negative impressions influenced the outcome of the game before the participants even started. While this is more aligned with the fear of vulnerability hypothesis, I'm not sure that you've shown that they don't also punish perceived slights.
5. In the trust game, how much does the pre-programmed partner actually betray the player, and is there randomness added to the response? If it's always benevolent and mirroring the player's responses, are they really getting betrayed, or is it just reciprocating the player's behavior? It seems harder to assess the punishment hypothesis when the partner never does anything worth being punished for, but perhaps I'm misunderstanding how the game was designed.
6. Measures of reciprocity seem inconsistent – in the methods, it's described as the factor of what was given (1.1-1.6), in some of the results it's described as the proportion of what was given back after the 3x multiplier (.3-.5), and in figure 5 it's the dollar amount. But all are labeled as "reciprocity". I found it confusing to keep track of what measure was being used in each analyses, and I think they each have different implications.
7. I am not convinced that your analysis adequately addressed the comparison of the higher punishment vs. lower risk-taking proposal. You claim that participants trust more after more reciprocity, so they aren't punishing the partner, and that the partner is programmed to positively increase after trusting behavior from the player, so you can't test punishment for not giving enough. But they also seem to trust less after less reciprocity, so couldn't that be punishment? Or is there another way to test that the partner didn't give as much as the player expected?
8. I would have thought you would compare decreased reciprocity vs increased reciprocity to test the punishment vs. vulnerability idea, but you only include that in the computational model, but not more basic analyses. One major example of this is figure 3B, in which you show differences in correlation between changes in reciprocity and trust, but I would have assumed that this relationship was nonlinear based on your argument. What happens if you separate them? I'd also like to see the actual data included, rather than just fit lines.
9. When you are comparing change in reciprocity and change in trust or willingness to trust, is it the next trial (i.e. in response)? Otherwise, I am having trouble understanding how it is not just a demonstration that your partner did what you programmed it to do. Could you clarify?
10. There are key analyses that you mention in your results section and then again highlight in the discussion, but they aren't the figures you selected. Figure 4A-D in particular seem to demonstrate that the computer did what you told it to do, and that your model parameters fit behavior well. These are useful checks, but don't push forward the larger argument. Instead, your discussion brings up group comparisons and condition differences, but you don't show figures of those results (with the exception of figure 2). Even then, I would include lines and color the dots to show the overall results as well as each group's result in Figure 2C.
11. Figure 3A is very interesting, but I'd like it to be unpacked a bit more. I'm having trouble understanding how lonelier individuals trusted more, when the differences at high reciprocity indicates they gave a lot less. Are there group, condition, or loneliness-related differences in average reciprocity or total earnings?
12. In figure 5, wouldn't you want reciprocity and willingness to trust to be on the x-axis? Those are measures that happened in the game, and the ratings of fairness and trustworthiness happened after the game (if I'm understanding correctly), suggesting that they based these assessments on the behavior of the game. Regardless, these still seem to demonstrate that generally there was an interpretation that more reciprocal partners are fairer and trustworthy (i.e., the programming worked). It is interesting that those that are more willing to trust also are rate the partner as fairer (5D), going back to the argument that perhaps the player's own behavior influenced their perception of the partner. However, these figures do not that demonstrate that loneliness or paranoia, nor the impression manipulation, influenced participants' responses, which seems like a key part of the argument. You mention these things in the text, and they would be far more interesting for your figures.
13. A very minor comment about figures but be consistent with the meaning of colors in your figures.
14. In the 5th paragraph of the discussion ("On the other..."), You mention that lonelier individuals have higher levels of trust but only in the lower levels of reciprocity. What if instead lonelier or more paranoid individuals are not considering the partner's perspective as well? Is that a possible explanation for being more reactive to perceived slights (not receiving back as much as they anticipated)? I would think that maxing their investment and getting back half of the total (1.5 the investment) would maximize the both the player and partner's outcomes fairly, so is it really a betrayal to not continue to return more (so the partner would be getting less)? How might theory of mind play into the game and potentially differ in lonelier or more paranoid individuals? Or do the lonelier and more paranoid players believe they should receive a greater amount than the partner?
15. This meta-analysis of the trust game may be useful in discussing the paranoia-related findings in the context of psychosis (Prasannakumar et al., 2023, Psychological medicine) 10.1017/S0033291722002562. I'd also recommend a paper discussing a model of trustee behavior in BPD, which highlights the role of irritability in perceptions of fairness (Hula et al., 2018; PLoS Comp Biology) <https://doi.org/10.1371/journal.pcbi.1005935>. Social value orientation may be another theory that can help explain differences in expectations in outcomes (Murphy et al., 2011) <https://doi.org/10.2139/ssrn.1804189>.
16. Please add a limitations section to the discussion. It would be helpful to comment on the sample size, and in particular issues with looking at interactions of loneliness and paranoia, when it appears uncommon to get individuals who are paranoid but not lonely (although it seems like there are more people who are lonely but not paranoid).

EDITORIAL POLICIES

We ask that you ensure your manuscript complies with our editorial policies and reporting requirements.

To that end, we require revised manuscripts to be accompanied by two completed items: a reporting summary that collects information on study design and procedure, and an editorial policy checklist that verifies compliance with all required editorial policies

- <https://www.nature.com/documents/nr-reporting-summary.zip>>Nature Research Reporting Summary
- <https://www.nature.com/documents/nr-editorial-policy-checklist.pdf>>Editorial Policy Checklist

All points on the policy checklist must be addressed. Your revised manuscript can only be sent back to the referees if these checklists are completed and uploaded with the revision.

Notes: If you have submitted a Stage 1 Registered Report, Review, Primer, Comment, or Perspective you do not need to submit these forms. If you have already submitted these forms, you may disregard this request.

** Visit Nature Research's author and referees' website at <http://www.nature.com/authors>>www.nature.com/authors for information about policies, services and author benefits**

Communications Psychology is committed to improving transparency in authorship. As part of our efforts in this direction, we are now requesting that all authors identified as 'corresponding author' create and link their Open Researcher and Contributor Identifier (ORCID) with their account on the Manuscript Tracking

System prior to acceptance. ORCID helps the scientific community achieve unambiguous attribution of all scholarly contributions. You can create and link your ORCID from the home page of the Manuscript Tracking System by clicking on 'Modify my Springer Nature account' and following the instructions in the link below. Please also inform all co-authors that they can add their ORCID to their accounts and that they must do so prior to acceptance.
<https://www.springernature.com/gp/researchers/orcid/orcid-for-nature-research>

Version 1:

Decision Letter:

Dear Dr Bellucci,

Thank you for submitting your manuscript titled "Willingness to trust is reduced by loneliness and paranoia" to Communications Psychology. We have given the paper our careful consideration and find it of potential interest. However, due to certain shortcomings we are concerned that sending the current manuscript out to review could lead to unnecessary delays and quite possibly an undesirable outcome of the review process.

In particular, we require appropriate statistical reporting to support the conclusions in the study. This includes 1) reporting the model comparison results (BIC scores) in a table in the main manuscript, 2) statistically comparing the size of correlations where applicable, and 3) reporting 90%, instead of 89%, confidence intervals.

In addition, we ask you to 4) include a sensitivity analysis in which you report the power-by-effect size achieved in your sample and effect sizes from comparable studies in the literature, 5) adopt the section order as suggested by our editorial request table attached in the previous decision letter.

We would therefore like to invite you to revise your manuscript to address these concerns before we make a final determination on whether to send your manuscript for external review.

We shall hope to receive your revised version as soon as you are able to complete the suggested revisions. If something similar is published in the interim we will have to consider the impact it has on the novelty of a revised manuscript.

If you anticipate a delay of more than four weeks, please let us know. Should your manuscript be substantially delayed without notifying us in advance and your article is eventually published, the received date may be that of the revised, not the original, version.

We also ask that you ensure your manuscript complies with our editorial policies and reporting requirements.

To that end, we require revised manuscripts to be accompanied by two completed items: a reporting summary that collects information on study design and procedure, and an editorial policy checklist that verifies compliance with all required editorial policies.

- [Nature Research Reporting Summary](https://www.nature.com/documents/nr-reporting-summary.zip)
- [Editorial Policy Checklist](https://www.nature.com/documents/nr-editorial-policy-checklist.pdf)

All points on the policy checklist must be addressed. Your revised manuscript can only be sent to referees if these checklists are completed and uploaded with the revision.

If you are not interested in submitting a suitably revised manuscript in the future please let me know immediately so we can close your file. If you have any questions, please contact me.

Please use the link below when you are prepared to resubmit.
Link Redacted

Thank you for your interest in Communications Psychology.

Best regards,
Troy Lui

Troy Lui, PhD
Associate Editor
Communications Psychology

Version 2:

Decision Letter:

Dear Dr Bellucci,

Thank you for your patience during the peer-review process. Your manuscript titled "Willingness to trust is reduced by loneliness and paranoia" has now been seen by 2 reviewers, and I include their comments at the end of this message. They find your work of interest but raised some important points. We are interested in the possibility of publishing your study in Communications Psychology, but would like to consider your responses to these concerns and assess a revised manuscript before we make a final decision on publication.

We therefore invite you to revise and resubmit your manuscript, along with a point-by-point response to the reviewers. Please highlight all changes in the manuscript text file.

Editorially, we consider it critical that the following two concerns are thoroughly addressed in the revised manuscript. First, additional psychometric evidence must be provided to justify the use of average of repeated momentary measures of loneliness to infer levels of trait-like loneliness and to support the validity of the results. Second, we require evidence of the effect size of $d = .71$ in Fett et al. (2012) as it is unclear whether it is reported in the paper; the relatively low sensitivity (in relation to more subtle effects revealed in the literature) must be noted as a limitation.

I am attaching an Editorial Requests Table that details critical reporting requirements for the revised manuscript. Please attend to each item and ensure your manuscript is fully compliant. If your revised manuscript is not aligned with these requests on major issues, such as those concerning statistics, it may be returned to you for further revisions without re-review.

Please submit the following items:

- Revised manuscript
- Point-by-point response to the referees' comments
- Cover letter (as a separate document)
- <https://www.nature.com/documents/nr-reporting-summary.pdf>-Nature Research Reporting Summary
- Completed Editorial Request Table (attached).

via this link: Link Redacted.

Additional guidance is available in our style and formatting guide <https://www.nature.com/documents/commspsychol-style-formatting-guide-accept.pdf>-Communications Psychology formatting guide.

Best regards,

Troy Lui

Troy Lui, PhD
Associate Editor
Communications Psychology

REVIEWER REPORTS:

Reviewer #1 (Remarks to the Author):

As noted in my initial review, the manuscript examined the behavioural patterns and responses in loneliness and paranoia underlying social uncertainty, as captured by the investment game in a sample of 54 participants (29 participants with non-affective psychosis and 25 healthy controls). The authors are commended for providing a thorough response to my and another reviewer's feedback, particularly in clarifying the methodology and results. However, some conceptual and narrative issues remain unaddressed or inadequately addressed..

My feedback regarding the lack of a clear narrative on the functional relationship between loneliness, paranoia, and underlying social cognitive biases has not been adequately addressed. In this version of the manuscript, this relationship remained to be described superficially and imprecisely e.g., 'various mechanisms might explain these thinking patterns' (p. 4). There are instances of circular reasoning that need to be clarified and strengthened- 'For instance, their elevated sensitivity to partner vulnerability might make them more cautious and suspicious about others' goodwill. Paranoid thinking might further exacerbate this heightened vulnerability by enhancing suspiciousness about others' good intentions' (p. 4). I would expect concise and direct statements that clarify whether these mechanisms are shared between loneliness and paranoia or are unique to each. This clarification is essential to provide a solid foundation for the study design and interpretation of results—such as whether a lack of trust or reciprocity is an antecedent or consequence of loneliness and/or paranoia, and whether paranoia necessarily co-occurs with loneliness and associated biased behavioral patterns and responses. Addressing this will also help shape the discussion to better highlight the theoretical implications of the analyses.

There are also instances where coherence could be strengthened. An example is the statements "Aspects of social disconnection are pronounced in individuals with psychotic disorders and loneliness has been linked to central symptoms of psychosis, including social threat perception and anhedonia (i.e., lack of reward/pleasure from social interactions). However, the magnitude of the associations suggests that loneliness and psychosis are related, yet distinct concepts and while initial, mostly cross-sectional studies probed their interrelationship, little is known about how they impact social interactions. Hence, investigating how loneliness modulates social interactions in individuals with and without a psychotic disorder will help to address an important knowledge gap" (p. 3). Questions for clarification include: What specific "associations" are being referred to? How large is the magnitude of these associations? In what way do these associations imply that "loneliness and psychosis are related"? Why is it that loneliness modulates (but does not mediate) social interactions among people with and without a psychotic disorder? These statements may cause confusion for readers and blur the focus of the study, particularly regarding the role of paranoia. Clarifying these points would improve coherence and strengthen the overall argument.

Other comments are as follows:

- The use of a two-group design with a clinical sample and a healthy control sample appeared to be a merit, but it is not well justified in the Introduction.
- I'm pleased to read that GPTS was used in the current study. Could the authors offer additional details of the distribution of participants by the established clinical cutoffs of GPTS, as in Freeman et al. (2021)? 'These statistics would provide support for the statement that 'paranoia should be understood as a continuum from non-clinical to clinical states' (p. 20). However, the reference cited for GPTS is the work of Freeman et al. (2021), which developed and validated the Revised-GPTS. Please clarify if GPTS or Revised GPTS was used in this study.
- The 'context effect' (p. 5) of $d = 0.71$ was referenced in the power calculation. However, it is not confusing what this 'context effect' refers to and how it was parameterized in the power calculation. I checked the reference cited (Fett et al., 2012) but couldn't find a clue for the effect size of 0.71.
- The meaning of the statement 'Further, the descriptively stronger relationship in controls could suggest that in controls paranoia is more context-dependent, but less so in individuals with clinical levels of paranoia' (p. 21) is unclear. What does it mean by 'more context-dependent'? Does the stronger association in the healthy control than in patients be explained by the greater range of paranoia? The distribution of the GPTS scores would give a clue to making sense of the difference in strength of associations between the groups.

- Why is the case that 'the relationship between loneliness and paranoid thoughts in the control group suggests that reducing paranoid thinking could help to reduce loneliness, and that vice versa reducing loneliness may be beneficial for improving paranoid thinking in the general population' (p. 21). Does this also apply to the clinical group, although the association between loneliness and paranoia is weaker? The bidirectional relationship seems to pinpoint circular reasoning- this confusion also speaks to a lack of conceptual clarity of the relationship between loneliness and paranoia, as well as the feasibility of translational implications of treatment and interventions. The same logic issue applies to the statement 'However, it's worth noting that we do find a strong correlation between loneliness and paranoia and these two characteristics were more strongly associated in controls than patients—again pointing to the transdiagnostic nature of loneliness and, most importantly, its relationship with mental health issues in the general population' (p. 21). Some rewriting may help clarify the arguments presented here.

- I do not agree with the authors' response that ESM was used "to have more stable estimate of participants' loneliness and paranoia states". Rather, its strength lies in the temporal resolution of capturing fluctuations of state loneliness and paranoia. According to Figure 2C, I guess loneliness and paranoia were operationalized as average ratings across 7 days (which is not clearly stated in the Methods section). If GPTS was used I would suggest using the GPTS total score in the analyses to better reflect the trait-like individual differences in paranoia (as formulated in the Introduction section), instead of the average of the ESM ratings. Also for the measure of loneliness, there needs to be a strong justification of how the average rating of the single, direct item of momentary loneliness reflects trait-like loneliness- it is widely discussed in the literature that trait and state loneliness take on different psychological processes and meanings.

Reviewer #2 (Remarks to the Author):

I commend the authors on their efforts to revise the manuscript. I have a few additional comments.

1. How did the authors define loneliness and paranoia? It appears to be from the ESM data, but is it an average? What days were measured (before or after or around the behavioral testing?), and how frequently did participants respond to the ESM? Although it was only one question about loneliness, the references support this method, and it is benefited by having multiple assessments throughout the day. A little clarity on the data collection would be helpful, however.

2. What effect is the power analysis referencing?

3. At line 338 (page 13), I think a word is missing that is important for clarity: "This suggests that while less (lonely?) individuals might share little out of cautiousness but are ready to increase their trust upon signs of a partner's goodwill, lonelier individuals might be more suspicious, and their lower investments might more directly signal distrust."

4. The description of figure 3C doesn't seem to match what is in the image (it seems like trust is greater for high loneliness individuals for the neutral condition), although the text where 3C is cited is more specific. Could the figure description be a bit more specific too?

5. It seems to me that the figure description of 5A doesn't match what the figure shows. Both the figure description and the in-text reference mention positive and negative reciprocity, but that is not in the figure. I also find it confusing that the symbol tau is next to the word reciprocity, but represents a change in willingness to trust. In addition, the description uses a euphemism for willingness to trust ("manifest a stronger discrepancy"), which makes the meaning unclear. Could you please clarify here?

6. The legend for loneliness in figure 3E doesn't match the other legends in the figure. Could this be changed to be in-line with the other sections?

7. Please describe what tau is in the figure description if not in the figure for 4D.

* TRANSPARENT PEER REVIEW: Communications Psychology uses a transparent peer review system. This means that we publish the editorial decision letters including Reviewers' comments to the authors and the author rebuttal letters online as a supplementary peer review file. However, on author request, confidential information and data can be removed from the published reviewer reports and rebuttal letters prior to publication. If your manuscript has been previously reviewed at another journal, those Reviewers' comments would not form part of the published peer review file.

Communications Psychology is committed to improving transparency in authorship. As part of our efforts in this direction, we are now requesting that all authors identified as 'corresponding author' create and link their Open Researcher and Contributor Identifier (ORCID) with their account on the Manuscript Tracking System prior to acceptance. ORCID helps the scientific community achieve unambiguous attribution of all scholarly contributions. You can create and link your ORCID from the home page of the Manuscript Tracking System by clicking on 'Modify my Springer Nature account' and following the instructions in the link below. Please also inform all co-authors that they can add their ORCIDs to their accounts and that they must do so prior to acceptance.
<https://www.springernature.com/gp/researchers/orcid/orcid-for-nature-research>

Version 3:

Decision Letter:

Dear Dr Bellucci,

Your manuscript titled "Willingness to trust is reduced by loneliness and paranoia" has now been seen by our reviewers, whose comments appear below. In light of their advice I am delighted to say that we are happy, in principle, to publish a suitably revised version in Communications Psychology.

We therefore invite you to revise your paper one last time to address the remaining concerns of our reviewers and a list of editorial requests. At the same time we ask that you edit your manuscript to comply with our format requirements and to maximise the accessibility and therefore the impact of your work.

EDITORIAL REQUESTS:

Please address reviewer #2's remaining concerns such as the assumptions of the statistical testing and the definition of vulnerability.

SUBMISSION INFORMATION:

OPEN ACCESS:

* DATA AVAILABILITY:

Link Redacted

Best regards,

Troby Lui

Troby Lui, PhD
Associate Editor
Communications Psychology

REVIEWERS' COMMENTS:

Reviewer #1 (Remarks to the Author):

I appreciate the authors' time and effort in providing a thorough response to my feedback and that of another reviewer, particularly in clarifying the theoretical relationship between loneliness and paranoia, as well as in discussing the interpretations of findings and the limitations of the current study. I have two minor comments for the authors' further consideration, as outlined below:

1. In line 69 the authors described the finding of the study by Lamster et al. (2017), which is a cross-sectional study modelling the structural relationship between loneliness and paranoia. The statement that "reducing loneliness leads to a decrease in paranoid symptoms" should be softened, for example: "reducing loneliness may lead to a decrease in paranoid symptoms."

2. In line 182, the decimal points in Spearman's correlations are missing.

The rest of the manuscript looks good to me. This work will advance understanding of social-cognitive biases underlying loneliness and paranoia in clinical and non-clinical populations.

Reviewer #2 (Remarks to the Author):

Thank you for your responses. I have a few more comments on the new sections you added. In addition, not all of my previous comments were addressed.

1. In the introduction, the word "vulnerability" gets used in what seems to be three different meanings (vulnerability to illness, a partner's vulnerability, and one's own vulnerability). In particular, I think the use of partner vulnerability was confusing in the context of the rest of the paper. Could you please clarify?

2. Could you please describe how often participants actually responded to the ESM? On average, did you get a 100% response rate, for instance? Were there any people excluded for not providing enough ESM responses?

3. For the ESM measures, you added some discussion of the group differences and variability of the measures, but it doesn't include statistical analysis to support your claims. I also worry that if the variance is different between the two groups, this would violate the assumptions of a t-test. It's most notable for the GPTS, which you are not really using in the rest of your analyses, but if you want to include these validations of the measures, you will benefit from using a non-parametric test (assuming the group variances are, in fact, different).

4. I think there is a disconnect between the story you are presenting in your figures and in the text. In particular, your discussion and conclusions highlight that the interaction of loneliness and paranoia indicate a particular sensitivity to partner reciprocity and fear of vulnerability, but this isn't presented in any of your figures directly. In general, paranoia is a small part of the story in the figures, which surprised me given the title. As it stands, the language around the interaction effect of loneliness and paranoia should be more reserved, as the sample size is low enough to make it difficult to test the effect.

5. I don't think you addressed my concern about the terms negative and positive reciprocity in Figure 5A. What is the benefit of including these terms if you are not showing them?

6. You did not describe tau in the caption of figure 4D.

Reviewers' Comments

Reviewer #1 (Remarks to the Author):

The manuscript examined the behavioural patterns and responses in loneliness and paranoia underlying social uncertainty, as captured by the investment game. The sample consisted of 54 participants (29 participants with non-affective psychosis and 25 healthy controls). The use of computational modelling revealed interesting behavioural tendencies (e.g. trust) in responses to variations in payoffs and expectations of players' responses (i.e., players were told to be benevolent or malevolent, or without this piece of information), which were found to be associated with individual differences in loneliness and paranoia. The experiment (i.e., investment) was sophisticatedly designed, the inclusion of both clinical and non-clinical participants in this sample, the use of an advanced modelling approach are merits of this study. These findings suggest biased social-cognitive processing belief formation and updating may underlie the perpetuation of loneliness and paranoia.

We thank the reviewer for this appreciation of our work.

My comments are as follows:

Major comment:

The manuscript would benefit from the improvement of theoretical clarity guiding this investigation. The bidirectional relationship between loneliness and paranoia is theorized and expected, while more studies, including experimental studies using loneliness manipulation and longitudinal studies, support loneliness as an antecedent of paranoia. The Introduction mentions several biased social-cognitive processes, lack of social motivation, lower trust and negative social expectation, but how these processes influence or are influenced by loneliness and paranoia is not entirely clear. Some descriptions are too vague (e.g., "various mechanisms might explain these thinking patterns", 4th paragraph, Introduction), that whether these mechanisms are shared between loneliness and paranoia, or unique to loneliness or paranoia warrants further clarification. I'm also confused about how these processes were represented or operationalized in the investment game paradigm.

Reply 1:

We thank the reviewer for pointing this out. We provided a brief overview of some of the cognitive biases in loneliness and paranoia but then focused on biased social learning and social inference mechanisms. We aimed to further clarify these mechanisms in the revised Introduction and Results section. For instance, on page 4:

"Various mechanisms might explain these thinking patterns. For instance, previous work has suggested a biased social learning mechanism in lonely individuals where lonelier individuals integrate more negative than positive information about others' social behaviors⁴³. This downweighing of positive information and beliefs about others could lead to similarly biased behavioral patterns. It is, thus, still unclear whether lonelier individuals form inaccurate social beliefs due to suboptimal learning or are simply reluctant to rely on positive beliefs about others²⁷. For instance, their elevated sensitivity to partner vulnerability might make them more cautious and suspicious about others' goodwill. Paranoid thinking might further exacerbate this heightened vulnerability by enhancing suspiciousness about others' good intentions¹⁹."

And on page 16:

"Differences in trusting behaviors could be explained by at least two different mechanisms: participants might be biased in updating their expectations of the partner's reciprocity, or they might accurately learn the partner's reciprocity but differently weight those expectations. To disentangle these routes, and to understand the inner working of how partner's behavior

shapes participants' choices, we developed a formal model for participants' learning expectations from the partner and weighing those expectations into their decisions. We also estimated participants' initial expectations of partner reciprocity, to investigate biases in those initial expectations and how they were updated based on reciprocities observed later."

In particular, loneliness and paranoia are distinct but interrelated concepts, the conceptual clarity should be carefully expressed. Meanwhile, as mentioned in the 1st paragraph of the Introduction, loneliness is believed to be a transdiagnostic phenomenon across psychopathologies, such as social anhedonia, depression and social anxiety, as well as other psychotic experiences beyond paranoia. Therefore, we expect lonely individuals do not necessarily have paranoid thinking, so both concepts should not be taken parallel, from the perspective of psychopathology and individual differences. Some of the arguments highlight the research gap appear only in the Discussion, and I would expect them to appear earlier to set the stage in the Introduction for the readers' easier understanding.

Reply 2:

We agree with the reviewer that lonely individuals do not necessarily have paranoid thinking and patients with paranoia are not necessarily lonely and we now mention this explicitly in the revised Introduction (page 3).

"Aspects of social disconnection are pronounced in individuals with psychotic disorders and loneliness has been linked to central symptoms of psychosis, including social threat perception and anhedonia (i.e., lack of reward/pleasure from social interactions)¹². However, the magnitude of the associations suggest that loneliness and psychosis are related, yet distinct concepts and while initial, mostly cross-sectional studies probed their interrelationship, little is known about how they impact social interactions. Hence, investigating how loneliness modulates social interactions in individuals with and without a psychotic disorder will help to address an important knowledge gap."

However, it's worth noting that we do find a strong correlation between loneliness and paranoia and these two characteristics were more strongly associated in controls than patients—again pointing to the transdiagnostic nature of loneliness and, most importantly, its relationship with mental health issues in the general population. We have now highlighted these points in the Discussion (page 17):

"In our study, we observed a strong relationship between loneliness and paranoia in individuals with and without non-affective psychosis. This finding is in line with the idea that paranoia should be understood as a continuum from non-clinical to clinical states, and that feelings of loneliness might induce a spiral of negative thoughts and maladaptive cognitions that could prelude and foster clinical symptoms^{26,53-55}. Further, the descriptively stronger relationship in controls could suggest that in controls paranoia is more context-dependent, but less so in individuals with clinical levels of paranoia⁵⁹. Similarly, the relationship between loneliness and paranoid thoughts in the control group stresses the importance of tackling loneliness to improve overall mental health. In line with this, a recent temporal analysis between loneliness and paranoia supports the hypothesis that reducing loneliness decreases paranoid thinking over time²⁹, but not vice versa. Similarly, preliminary experimental evidence in a questionnaire-based study also suggests that experimentally decreasing loneliness leads to a reduction of paranoia⁵⁹."

Other comments:

Abstract:

- The abstract should also briefly mention the study design, including the sample and the behavioural task.

Reply 3:

We thank the reviewer for this suggestion and updated the abstract accordingly.

“Here, we investigated the relationship between paranoid thoughts and lonely individuals’ willingness to rely on expectations of partner reciprocity in an investment game with individuals with and without psychosis (54 participants).”

Introduction:

- The Introduction should end with brief descriptions of the experimental design, the behavioral paradigm used, key variables/ parameters to look at, and operationalization of the variables, the use of both clinical and non-clinical samples, and also studies hypotheses. At the current stage, the linkage between the Introduction and Results needs to be strengthened, as the Methods only appear at the end of the manuscript and you should expect the readers could grasp the core of the study design when they read the Results right after the Introduction.

Reply 4:

We have now moved the Methods before the Results, following the journal’s guidelines. Now, the reader will find explanations to all variables before the Results. In particular, at the beginning of the Methods, the reader will see a “Task” subparagraph that provides the following explanations (page 6):

“To measure trust, a modified multi-round investment game was used (**Fig. 1**)^{30,37}. In this neuroeconomic computer-based game, consisting of 20 trials each, participants played the role of the investor. They were instructed that they played with three different human counterparts in another location via the Internet. In fact, they were playing against a computer that was pre-programmed to behave in a probabilistic and benevolent manner. Trustees were pre-programmed to play an overall benevolent, tit-for-tat strategy. The first repayment (i.e., reciprocity) was either 1.0, 1.1, 1.2, 1.3, 1.4 or 1.5 times (i.e., repayment rates) the first investment (i.e., the amount shared by the participant and indicative of their trust), with an equal chance for returns of each repayment rate. After the first trial, the repayment rates were updated depending on the amount being invested compared to the amount invested in the previous trial (i.e., trust change: $\delta_t = trust_t - trust_{t-1}$). When the investor’s current trust increased or when participants continued to invest the maximum amount (£10), then an increment of 0.05 was added to each of the randomly selected repayment rates until each of the factors reached 1.6, thus reflecting higher reciprocity. Each repayment rate decreased with the same increments (-0.05) when trust decreased or remained at the minimum (£0), with a minimum value of 1 for each repayment rate, thus reflecting lower reciprocity. This way of programming ensured reciprocity that would resemble subtle changes in trust. Moreover, since repayment rates were chosen probabilistically between the above range, they still allowed for trials in which trustees manifested either positive (increases of) or negative (decreases of) reciprocity (i.e., trustees were not deterministically mimicking participants’ strategy). This design allowed us to investigate participants’ reactions to positive and negative reciprocity.”

Moreover, we followed the reviewer’s suggestion and at the end of the Introduction, we briefly mentioned the paradigm used, the sample, the methodology, and the study hypotheses as follows:

“Here, we investigated the relationships of loneliness and paranoia with belief formation and trust in a sample of participants with and without a psychotic disorder. Combining a modified investment game (**Fig. 1**) with Bayesian computational modeling, we investigated whether lonely individuals responded to their partner’s reciprocity by showing a differential weighting of their expectations. We expect that if lonely individuals differentially weighted their expectations of partner reciprocity based on the observed behavioral feedback, they would show a different sensitivity to changes in partner reciprocity. For instance, they might be more

cautious after signs of decreasing cooperation but more generous after signs of increasing cooperation. Further, we tested the relationship of paranoid thinking with expectation formation and weighting in lonely individuals. We hypothesized that higher levels of paranoid thoughts might bias lonely individuals' willingness to rely on their expectations of partner reciprocity due to higher perceived vulnerability, leading to a decrease in trust”

Methods and Results:

- It's not clear how the variables for analysis were measured and matched with the variables used in the analysis (e.g. , reciprocity, trust). As there are quite many parameters (e.g., ratings of trustworthiness, the amount invested in each trial), it is not straightforward to see how these parameters were treated and matched with variables for analysis. Maybe a table listing these explaining the variables and parameters would facilitate comprehension.

Reply 5:

We thank the reviewer for pointing to this. We aimed to improve clarity in three ways. Instead of a table listing all the variables, we now provide a simpler visual representation of the task procedures and variables measured (Fig. 1). Moreover, we have revised the description in the text to enhance clarity regarding each of the variables.

First, we described how we operationalized the measures in the caption of the task figure (Fig. 1) as follows:

“At the beginning of the investment game, participants' (investor) expectations of their partner's (trustee) returns were manipulated by making them believe that their partner was returning their trust generously (positive condition) or not (negative condition); in the neutral condition, no expectation manipulation took place (neutral condition). Participants played 20 trials in each condition (within subject) and the amount they shared in the game (investment) measured their trialwise trust. At the end of the game, they were asked to rate the trustworthiness, fairness, and generosity of each of the partners in that game (on a 7-point Likert scale).”

Second, in the Methods (now before the Results section) we described the operationalization of trust and reciprocity in the task (page 6):

“The first repayment (i.e., reciprocity) was either 1.0, 1.1, 1.2, 1.3, 1.4 or 1.5 times (i.e., repayment rates) the first investment (i.e., the amount shared by the participant and indicative of their trust), with an equal chance for returns of each repayment rate. After the first trial, the repayment rates were updated depending on the amount being invested compared to the amount invested in the previous trial (i.e., trust change: $\delta_t = trust_t - trust_{t-1}$).”

and the operationalization of participants' impressions of their partner (page 7):

“At the end of the experiment, we asked participants to rate their partners' trustworthiness, fairness, and generosity on a 7-point Likert scale to obtain a measure of participants' overall impressions of their partners' social behavior during the investment game.”

Finally, we also clarified in the Methods how the different parameters were used in the analyses (page 7):

“A mixed-effect Bayesian model was implemented to predict trialwise trust (amount shared by participants) with group, experimental condition and trialwise reciprocity (amount shared back by the algorithm) as regressors”

And for ratings as follows (page 7-8):

“Similar Bayesian regression models with group and experimental conditions as regressors were run to predict post-task impression judgments (i.e., ratings of fairness, generosity and trustworthiness on a 7-point Likert scale).”

- Was there any a-prior sample size calculation?

Reply 6:

We confirm that we ran an a-priori sample size calculation to analyze context effects. Our initial sample size considerations of N = 50 were based on d = .71 for a previously established context effect (Fett et al. 2012), and calculated for an independent sample t-test, with a desired power of .80. We have now reported our sample size computations in the manuscript (page 5) and our response to the reviewers.

- It's interesting to see that participants were explicitly told about the interacting partners' predicted behavioural tendencies (i.e., to be benevolent or malevolent) before the trials. How would this experimental manipulation interact with the probabilistic payoffs in shaping participants' investment decisions? Cognitive models of paranoia/ persecutory delusions have posited the unique role of negative-others schemas, as well as hostility attribution bias, in the development and maintenance of paranoia. It should be expected when engaging in the experiments, participants were primed about the hostility of the players, which may further activate negative-other schemas to trigger mistrust or paranoid thinking (and indeed participants with high paranoia may be less willing to buy in the information that the players were with good intention). How would this (and its impacts on behaviors such as trust and belief formation/ updating, either intended or unintended), be modelled in the analysis?

Reply 6:

We agree with the reviewer and this is indeed what we see in our results. In particular, patients' negative schemata should have impacted their expectations and behaviors particularly in the positive condition, as those schemata would have acted against our attempt to make participants more positively inclined towards their partners in the game. Different pieces of evidence support this reasoning.

First, we saw that healthy controls were more likely to increase their trust in the positive condition than patients (page 12). Second, we observed that patients had more negative prior expectations in the neutral condition as compared to controls (p.16). despite all having played with benevolent trustees whose behavior did not differ across groups and conditions (as checked on page 12). Finally, the impact of negative schemata on both behavior and expectations is captured by the τ parameter and expectations ρ , respectively (see Methods and Fig. 4A).

- I suppose the authors consider loneliness and paranoia as more trait-like aspects of these phenomena in their conceptualization of research questions (e.g. individuals who are more lonely/ paranoid were hypothesized to show less trust when they expected their partners to cheat or not to reciprocate). However, well-established measures of loneliness (e.g. UCLA Loneliness Scale) and paranoia (e.g. Green et al. Paranoid Thoughts Scale) were not mentioned (and maybe not administered?) throughout the manuscript. Rather, state-like, momentary measures of loneliness and paranoia were captured with the experience sampling method. I guess loneliness and paranoia in this study were conceptualized with the mean ratings of the experience sampling measures of both phenomena respectively (please confirm!). If so, there is a disjoint between theory/ research questions and the measures of loneliness and paranoia. The inclusion of well-validated measures of loneliness and paranoia is important, as these are the main variables of interest. Please clarify this methodological issue.

Reply 7:

We are happy to clarify this. We used ESM to have more stable estimate of participants' loneliness and paranoia states. For loneliness, we used the single-item measure as previously suggested

(https://www.ons.gov.uk/peoplepopulationandcommunity/wellbeing/compendium/nationalmeasureofloneliness/2018/recommendednationalindicatorsof_loneliness). We mention this in the Methods on page 7 as follows:

“Loneliness was assessed with the item “I feel lonely”, as previously done²¹ and currently recommended by the Office of National Statistics⁶⁸.”

For paranoia, we collected paranoia measures using both ESM and the Green et al. Paranoid Thoughts Scale (the scale mentioned by the reviewer) as a well-established control measure of paranoia. We now have added information on the GPTS to the revised manuscript. ESM paranoia and paranoia, as well as ideas of reference assessed with the GPTS were highly correlated, thereby supporting the construct validity of our ESM measure of paranoia. We reported this in the manuscript on page 7 as follows:

“Paranoia was assessed for every participant based on an average of the following items: “I feel suspicious”; “I feel safe (reverse scored)”; “I feel others dislike me”; and “I feel others intend to harm me”. Moreover, we also collected paranoia scores using the Green et al. Paranoid Thoughts Scale (GPTS)¹, as a well-established control measure of paranoia to validate our ESM measure of paranoia. Our ESM measure of paranoia was highly correlated with GPTS score for both GPTS subscales (reference: $\rho_{52} = .62$, $p < .0001$; persecution: $\rho_{52} = .74$, $p < .0001$), indicating high validity of our ESM measure.”

Discussion:

- In the second paragraph of the Discussion, the authors suggest ‘the strong relationship between loneliness and paranoia in our participants without a psychotic disorder might point to the importance of tackling loneliness to alleviate paranoid thinking and vice versa’, the logic of the argument didn’t read well. Please rewrite to improve coherence and logic. Indeed, the directionality of the relationship between loneliness and paranoia can't be tested in this study.

Reply 8:

We have rewritten the paragraph to clarify and stress the importance between loneliness and paranoia, and the necessity to conduct more studies investigating the directionality of this relationship. The paragraph now reads (page 21):

“Similarly, the relationship between loneliness and paranoid thoughts in the control group suggests that reducing paranoid thinking could help to reduce loneliness, and that vice versa reducing loneliness may be beneficial for improving paranoid thinking in the general population. The current findings cannot speak to the directionality of the relationship, however a recent temporal analysis between loneliness and paranoia supports the hypothesis that reducing loneliness decreases paranoid thinking over time²⁹, but not vice versa. Similarly, preliminary experimental evidence also suggests that experimentally decreasing loneliness leads to a reduction of paranoia⁵⁹. Further studies are needed to investigate the causal and temporal mechanisms underlying the loneliness-paranoia association.”

Overall:

- Please avoid using casual language such as “xxx affect xxx”, “the impact of...” and “the effect of...”, as causal conclusions from this design are limited.

Reply 9:

We thank the reviewer for this suggestion and revised the manuscript accordingly to avoid causal language.

Reviewer #2 (Remarks to the Author):

This manuscript examines the influence of loneliness and paranoia on trust behaviors as the investor in an iterated trust game in healthy controls and patients with psychosis. They additionally applied a computational vulnerability model to measure concerns of vulnerability (willingness to trust) and asymmetrical responses to a benevolent partner's reciprocity. The stated aim to provide insight into how loneliness and paranoia influence negative impression formation; however, there are several ideas presented that do not appear cohesive throughout the manuscript, although I am not sure if this is due to inconsistency in ideas or writing style. The overarching aim of the study is interesting and valuable to understanding distrust and factors that may contribute to its development, but I have several concerns. I am not convinced that the current analyses support the claims made. Below are some concerns with the framing of the current document as well as thoughts to consider for a future manuscript.

We thank the reviewer for this evaluation of our work. We have addressed the reviewer's concerns as outlined below. As part of the revisions, we ran additional analyses and improved the figures to visualize the effects of interest.

1. In order to put the methods at the end, more needs to be explained in simple terms in the introduction and results sections. It doesn't need to be all the details, but enough that a person can understand the game, the manipulation, how many participants were included, and the definitions of independent and dependent variables.

Reply 1:

We thank the reviewer for pointing to this. Following the journal's guidelines, we have now moved to Methods before the Results, so that all details related to the sample, design, and variables are clarified before the Results section (pages 5-6):

2. With that said, I don't think the vulnerability model needs to be explained in the results so thoroughly; instead define the variables enough so a reader knows what they represent, and then the math can be in the methods. Stay consistent with how you refer to them as well, as sometimes they are referred to as phrases and sometimes, they are referred to as characters. My personal preference is to use a word or phrase with the character in parentheses (i.e., willingness to trust τ). In addition, I would like more explanation for what action probabilities represent in the model.

Reply 2:

We have removed the exposition of the model from the Results section. The model is now explained only in the Methods now coming before the Results (pages 8-9). Further, we explained what action probabilities represent on page 18:

"action probability (i.e., the utility-based probability of choosing a given investment)"

3. The 4th paragraph of introduction is unclear to me in its current form. From what I understand, you are comparing hypotheses to explain suboptimal behavior in lonely individuals: increased punishment vs. fear of vulnerability. Conceptually that makes sense, but the wording took a while to figure out what was meant, and it felt unclear if or how these hypotheses were being directly compared.

Reply 3:

We apologize for the lack of clarity. We rewrote the paragraph to highlight the different hypotheses that outline that biased social behaviors in lonely individuals might be due to suboptimal social learning or reluctance to rely on positive beliefs about others. We revised as follows on page 4:

“Various mechanisms might explain these thinking patterns. For instance, previous work has suggested a biased social learning mechanism in lonely individuals, where lonelier individuals integrate more negative than positive information about others’ social behaviors⁴³. This downweighing of positive information and beliefs about others could also lead to similarly biased behavioral patterns. It is, thus, still unclear whether lonelier individuals form inaccurate social beliefs due to suboptimal learning or are simply reluctant to rely on positive beliefs about others²⁷. For instance, their elevated sensitivity to partner vulnerability might make them more cautious and suspicious about others’ goodwill. Paranoid thinking might further exacerbate this heightened vulnerability by enhancing suspiciousness about others’ good intentions¹⁹.”

4. I think the fear of vulnerability hypothesis fits with the end of the results & discussion section, which is that lonelier & more paranoid individuals give less initially and don’t respond as strongly to positive reciprocity from the partner, but it wasn’t clear how it fit in the larger context of the paper due to differences in language use. The authors argue that this behavior leads to lower reciprocity from the partner (as it’s programmed to do), causing a feedback loop. For what it’s worth, I think this paragraph is stronger. In particular, showing that prior expectations of reciprocity was associated with fairness impressions and willingness to trust in the game suggests that negative impressions influenced the outcome of the game before the participants even started. While this is more aligned with the fear of vulnerability hypothesis, I’m not sure that you’ve shown that they don’t also punish perceived slights.

Reply 4:

We thank the reviewer for pointing to this. We agree that showing that prior expectations of reciprocity were associated with fairness impressions and willingness to trust in the game suggests that negative impressions influenced the outcome of the game before the participants even started, which is more aligned with the fear of vulnerability hypothesis. In our revisions, we further aimed to clarify which behavioral patterns would suggest support for the vulnerability vs punishment hypotheses (pages 13-14).

“Stronger reductions in trust in lonelier individuals could be explained by at least two mechanisms: a punitive mechanism by which lonely individuals might have wanted to punish their partner for being uncooperative and unkind; or a protective mechanism by which lonely individuals might have feared a betrayal and hence pre-emptively reduced their vulnerability to the partner. A punitive mechanism might be triggered if a trustee responds with an inadequate level of reciprocity to a participant’s sign of cooperation (i.e., increase of trust). As a consequence, a decrease of trust in response to positive reciprocity could signal punitive behavior. However, as shown above, despite cases of breaches of trust, the trustees were on average reciprocating participants’ increases of trust with positive reciprocity and overall, participants’ trust changes positively correlated with changes in partner reciprocity ($\rho_{224} = .18$, $p = .0083$), suggesting that participants were seeing their trust honored by their partners and were responding by increasing their own trust.

On the contrary, a vulnerability mechanism could be triggered by the fear of experiencing breaches of trust (i.e., negative reciprocity). This fear should be greater in those interactions where there was more at stake, such as interactions with highly-reciprocating partners whose negative reciprocity meant greater losses. Indeed, interactions with highly-reciprocating partners involved higher levels of trust ($\rho_{224} = .32$, $p < .00001$) and since we pre-programmed all partners not to share more than half of what they received (a fairness upper bound), highly-reciprocating partners were more likely to show negative reciprocity in future trials ($\rho_{224} = -.80$, $p < .00001$). This, together with lonely individuals’ lower trust levels and weaker sensitivity

to positive reciprocity in interactions with highly-reciprocating trustees, suggests that lonely individuals' trust was more strongly modulated by a fear of being vulnerable to a partner's betrayal. This could also explain why lonelier individuals showed patterns of trusting behaviors that more closely resembled an "all-or-nothing" strategy (Fig. 3E). They might have shared high amounts to manifest strong cooperation but, as the partner increased their level of reciprocity (Fig. 3A), they might have feared a betrayal and either completely withdrew from the interaction (i.e., sharing nothing) or reduced their trust to the lowest levels (i.e., sharing the smallest amounts) as to probe the partner's "true" reciprocity (Fig. 3E)."

5. In the trust game, how much does the pre-programmed partner actually betray the player, and is there randomness added to the response? If it's always benevolent and mirroring the player's responses, are they really getting betrayed, or is it just reciprocating the player's behavior? It seems harder to assess the punishment hypothesis when the partner never does anything worth being punished for, but perhaps I'm misunderstanding how the game was designed.

Reply 5:

We thank the reviewer for this comment and the opportunity to clarify. Indeed, the trustees were pre-programmed to play an overall benevolent, tit-for-tat strategy. In particular, trustees were pre-programmed to share between 1 and 1.5 of what they received from the participant/investor (i.e., repayment rates). These repayment rates lead to reciprocity levels classically considered "fair" in the literature (as they end up producing returns between one third and half of the trustee's endowment).

Moreover, these repayment rates were updated depending on participants' changes in trust (investments) over investment game trials. In particular, an increment of 0.05 was added to each of the randomly selected factors until each of the factors reached 1.6 after an increase of trust, so that trustees were more likely to increase reciprocity after an increase of trust from one game trial to the next. Similarly, each factor decreased with the same increments (-0.05) when current investments reflected a decrease of trust from the previous investment or remained at the minimum (£0), leading trustees to be less reciprocal. Importantly, though, repayment rates were chosen probabilistically between the above range, so that trustees could still manifest positive reciprocity after a decrease of trust and negative reciprocity after an increase of trust (which constituted examples of breaches of trust in the game). We clarified this in the Methods as follows on page 6:

"Trustees were pre-programmed to play an overall benevolent, tit-for-tat strategy. The first repayment (i.e., reciprocity) was either 1.0, 1.1, 1.2, 1.3, 1.4 or 1.5 times (i.e., repayment rates) the first investment (i.e., the amount shared by the participant and indicative of their trust), with an equal chance for returns of each repayment rate. After the first trial, the repayment rates were updated depending on the amount being invested compared to the amount invested in the previous trial (i.e., trust change: $\delta_t = trust_t - trust_{t-1}$). When the investor's current trust increased or when participants continued to invest the maximum amount (£10), then an increment of 0.05 was added to each of the randomly selected repayment rates until each of the factors reached 1.6, thus reflecting higher reciprocity. Each repayment rate decreased with the same increments (-0.05) when trust decreased or remained at the minimum (£0), with a minimum value of 1 for each repayment rate, thus reflecting lower reciprocity. This way of programming ensured reciprocity that would resemble subtle changes in trust. Moreover, since repayment rates were chosen probabilistically between the above range, they still allowed for trials in which trustees manifested either positive (increases of) or negative (decreases of) reciprocity (i.e., trustees were not deterministically mimicking participants' strategy). This design allowed us to investigate participants' reactions to positive and negative reciprocity."

6. Measures of reciprocity seem inconsistent – in the methods, it’s described as the factor of what was given (1.1-1.6), in some of the results it’s described as the proportion of what was given back after the 3x multiplier (.3-.5), and in figure 5 it’s the dollar amount. But all are labeled as “reciprocity”. I found it confusing to keep track of what measure was being used in each analyses, and I think they each have different implications.

Reply 6:

We apologize for the lack of clarity. The factors refer to the trustee’s repayment rates which changed in response to changes in investor trust (i.e. shared amount in the investment game). We have now consistently labeled them as “repayment rates” in the Methods and at the beginning of the Results where we explained the algorithm. Reciprocity refers to what the trustee shared back as the fraction (proportion) of what they received from the investor. Except for when we explained the underlying algorithm, we always referred to reciprocity as the proportion of what has been received from the investor throughout the manuscript.

Notably, repayment rates and reciprocity are closely related to each other. Let’s denote the current reciprocity as r_t , the current repayment (i.e., amount shared back by the trustee) as a_t , the current investment (i.e., amount shared by the participant) as s_t , the multiplication factor as μ and the current repayment rate as η_t , we have:

$$r_t = \frac{a_t}{\mu s_t} = \frac{\eta_t s_t}{\mu s_t} = \frac{\eta_t}{\mu}$$

Hence, the repayment rate is related to reciprocity as a fraction of the multiplication factor. Moreover, this also means:

$$a_t = \mu s_t r_t$$

where $\mu = 3$, which is how we defined repayment in Fig. 4a.

7. I am not convinced that your analysis adequately addressed the comparison of the higher punishment vs. lower risk-taking proposal. You claim that participants trust more after more reciprocity, so they aren’t punishing the partner, and that the partner is programmed to positively increase after trusting behavior from the player, so you can’t test punishment for not giving enough. But they also seem to trust less after less reciprocity, so couldn’t that be punishment? Or is there another way to test that the partner didn’t give as much as the player expected?

Reply 7:

We apologize for the lack of clarity and we have now revised the Results section to clarify the difference between three types of behavioral patterns. We believe we can best answer the reviewer’s questions by breaking them down into three points: 1) how did participants behave?; 2) how did the partner behave?; 3) how did lonely individuals behave?

- 1) It is true that participants trusted more in response to high levels of partner reciprocity, and yes, their trust changes strongly correlated with partner reciprocity changes. We mentioned this in the Results as follows (page 11):

“We first analyzed participants’ trusting behavior and their partners’ reciprocity in the game. We observed that participants shared more with reciprocating partners ($\beta = 8.51$, 90% CI = [6.28, 11.18]) and their trust was overall higher if their partner had higher levels of reciprocity in previous trials ($\beta = 5.22$, 90% CI = [1.69, 8.66]).”

On page 12:

“We observed that previous lower levels of trust were more likely to predict subsequent increases of trust in the positive condition ($\beta = -0.05$, 90% CI = [-0.10, -0.01]), suggesting that participants were slowly increasing their trust over trials and were more likely to do so when they thought their partner was benevolent.”

And also on page 13:

“overall, participants’ trust changes positively correlated with changes in partner reciprocity ($\rho_{224} = .18$, $p = .0083$)”

Moreover, we also defined what we mean by punishment on page 13:

“A punitive mechanism might be triggered if a trustee responds with an inadequate level of reciprocity to a participant’s sign of cooperation (i.e., increase of trust). As a consequence, a decrease of trust in response to an increase of reciprocity could signal punitive behavior”

This is in line with classic definitions of punishment in the investment game, as we clarify in the Discussion on page 22:

“refusing to share money in the investment game—classically interpreted as a self-interested choice—can also signal irritation and disrespect, and hence be used as a form of social punishment^{41,58}”

Given this evidence, we came to the same conclusion as the reviewer: participants were not punishing their partner, and we think this was important to clarify in the analysis in order to rule out the punishment hypothesis as possible explanation of participants’ behavior. Importantly, though, participants’ expectations played a role in how they reacted to their partner as evidenced by the condition effect.

- 2) **Further, we can test whether our partners showed behaviors that fell short of participants’ expectations of partner reciprocity. It is indeed true as the reviewer say that our partners were pre-programmed to show overall benevolent behavior. However, we also induced fluctuations in the trialwise reciprocity rate implemented in the game. This led to trials in which participants experienced negative reciprocity. We added an additional figure to visualize this and clarified it in the manuscript as follows (pages 12):**

“We also checked that trustees behaved as expected and in similar fashion across groups and conditions, so as to rule out the possibility that potential biases in participants’ behaviors were induced by our algorithm’s different responses. Indeed, we observed that our pre-programmed partners increased their reciprocity in response to increases of participants’ trust ($\beta = 7.33$, 90% CI = [4.78, 10.02]), and they did so in similar ways both across conditions ($\beta = -0.01$, 90% CI = [-0.16, 0.13]) and groups ($\beta = -0.06$, 90% CI = [-0.48, 0.39]). Moreover, on average, lonelier individuals did not experience lower levels of reciprocity ($r_{52} = 0.10$, $p = 0.495$) or earnings ($r_{52} = 0.13$, $p = 0.353$) from their partners. Importantly, even though we programmed our participants’ partners to be overall benevolent, their behavior constantly fluctuated over trials, leading to cases of negative reciprocity across the game (Fig. 3B).”

- 3) **On the contrary, we observed that lonelier individuals were less likely to increase their lower levels of trust in response to positive reciprocity. Given that positive reciprocity implies that the partner had honored participants’ trust, a**

failure to increase trust in response to this sign of “goodwill” cannot be explained by a punitive mechanism but is more likely to be explained by the vulnerability hypothesis. The fact that lonelier individuals were less likely to trust highly-reciprocating partners who were more likely to breach trust due to the implemented reciprocity upper bound suggests that they might have been more suspicious of their underlying motives and more sensitivity to future betrayal. That their lower increases of trust further correlated with individual levels of paranoia provides complementary evidence to the vulnerability hypothesis. We clarified this in the manuscript as follows (page 13):

“We observed that greater feelings of loneliness were associated with lower sensitivity to partner reciprocity, particularly in the negative condition ($\beta = -0.34$, 90% CI = [-0.54, -0.14]), suggesting that lonelier individuals reduced their trust in that condition more than in the other conditions (Fig. 3C). Importantly, lonely individuals’ smaller increases of trust were contingent on their previous trust levels (Fig. 3D). In particular, lonelier individuals were less likely to increase their trust had they manifested lower levels of trust in the previous trial, especially in the positive condition ($\beta = 0.05$, 90% CI = [0.01, 0.08]) and despite having experienced positive reciprocity from the partner ($\beta = 0.47$, 90% CI = [0.02, 0.90]). This suggests that while less individuals might share little out of cautiousness but are ready to increase their trust upon signs of a partner’s goodwill, lonelier individuals might be more suspicious, and their lower investments might more directly signal distrust.”

And again, on page 13-14:

“On the contrary, a vulnerability mechanism could be triggered by the fear of experiencing breaches of trust (i.e., negative reciprocity). This fear should be greater in those interactions where there was more at stake, such as interactions with highly-reciprocating partners whose negative reciprocity meant greater losses. Indeed, interactions with highly-reciprocating partners involved higher levels of trust ($\rho_{224} = .32$, $p < .00001$) and since we pre-programmed all partners not to share more than half of what they received (a fairness upper bound), highly-reciprocating partners were more likely to show lower reciprocity in future trials ($\rho_{224} = -.80$, $p < .00001$). This, together with lonely individuals’ lower trust levels and weaker sensitivity to positive reciprocity in interactions with highly-reciprocating trustees, suggests that lonely individuals’ trust was more strongly modulated by a fear of being vulnerable to a partner’s betrayal.”

8. I would have thought you would compare decreased reciprocity vs increased reciprocity to test the punishment vs. vulnerability idea, but you only include that in the computational model, but not more basic analyses. One major example of this is figure 3B, in which you show differences in correlation between changes in reciprocity and trust, but I would have assumed that this relationship was nonlinear based on your argument. What happens if you separate them? I’d also like to see the actual data included, rather than just fit lines.

Reply 8:

We apologize for the confusion. The regressor coefficient represented changes in reciprocity, both decreases and increases of reciprocity. We specified this in the Methods as follows (page 7):

“A similar model was implemented predicting trust changes (as the difference in investments between the current and the previous trial) and changes (increases/decreases) in partner reciprocity (as the difference in partner reciprocity between the last two trials) as regressors”

This means that a significant interaction could be interpreted as higher trust changes in response to higher reciprocity changes or lower trust changes in response to lower reciprocity changes. Testing the punishment and vulnerability hypothesis is hard with model-agnostic analyses but we provided tentative evidence in our regression analyses by investigating changes in trust as a function of changes in partner reciprocity and current trust levels, which indicate a participant's level of vulnerability, given that, as explained in the modeling analyses, investments can be psychologically thought of as a loss and hence representing the degree of a participant's vulnerability. We mentioned these model-agnostic analyses in Reply 7 to the reviewer and on page 13 as follows:

"In particular, lonelier individuals were less likely to increase their trust had they manifested lower levels of trust in the previous trial, especially in the positive condition ($\beta = 0.05$, 90% CI = [0.01, 0.08]) and despite having experienced positive reciprocity from the partner ($\beta = 0.47$, 90% CI = [0.02, 0.90])."

We indeed can have more complex relationships than the one that can be tested with regression models. Importantly, our computational modeling builds on these sequential effects to gain more in-depth insights into these dynamics. Finally, we have now removed the fitted lines and showed the overall effects by binning data across different predictor levels (see new Fig. 3D-C-E).

9. When you are comparing change in reciprocity and change in trust or willingness to trust, is it the next trial (i.e. in response)? Otherwise, I am having trouble understanding how it is not just a demonstration that your partner did what you programmed it to do. Could you clarify?

Reply 9:

We are happy to clarify this. Indeed, reciprocity and trust changes were always the difference in shares between two consecutive trials. We clarified this in the Results section (page 12):

"We then analyzed sequential and conditional effects to investigate the effects of positive and negative reciprocity (as the difference in partner reciprocity between the last two trials) on participants' trust changes (as the difference in investments between the current and the previous trial)."

And in the Methods (page 7):

"A similar model was implemented predicting trust changes (as the difference in investments between the current and the previous trial) and changes (increases/decreases) in partner reciprocity (as the difference in partner reciprocity between the last two trials) as regressors"

Relatedly, there are two different mechanisms that can be tested and this may have led to some confusion.

On one hand, we can investigate the responses of the *trustees* to changes in participants' trust. This would demonstrate that our trustees actually did what we programmed them to do. We ran analyses to test this and reported them in the Results. From these results, it is clear that 1) our trustees behaved as programmed; and most importantly, 2) that their behavior didn't differ between groups or depending on the degree of loneliness. This is important because it demonstrates that biases in participants' behavior could not be traced back to differences in their partner's behavior (as intended). We report these results as follows (page 12):

“We also checked that trustees behaved as expected and in similar fashion across groups and conditions, so as to rule out the possibility that potential biases in participants’ behaviors were induced by our algorithm’s different responses. Indeed, we observed that our pre-programmed partners increased their reciprocity in response to increases of participants’ trust ($\beta = 7.33$, 90% CI = [4.78, 10.02]), and they did so in similar ways both across conditions ($\beta = -0.01$, 90% CI = [-0.16, 0.13]) and groups ($\beta = -0.06$, 90% CI = [-0.48, 0.39]). Moreover, on average, lonelier individuals did not experience lower levels of reciprocity ($r_{52} = 0.10$, $p = 0.495$) or earnings ($r_{52} = 0.13$, $p = 0.353$) from their partners. Importantly, even though we programmed our participants’ partners to be overall benevolent, their behavior constantly fluctuated over trials, leading to cases of negative reciprocity across the game (Fig. 3B).”

On the other hand, we can investigate the responses of our participants (i.e. investors) to changes in trustee’s reciprocity. This demonstrates how differently our participants reacted to the observed levels of reciprocity from the trustee partner (page 12).

“We then analyzed sequential and conditional effects to investigate the effects of positive and negative reciprocity (as the difference in partner reciprocity between the last two trials) on participants’ trust changes (as the difference in investments between the current and the previous trial). We observed that previous lower levels of trust were more likely to predict subsequent increases of trust in the positive condition ($\beta = -0.05$, 90% CI = [-0.10, -0.01]), suggesting that participants were slowly increasing their trust over trials and were more likely to do so when they thought their partner was benevolent. Moreover, healthy controls were more likely to increase their trust in the positive condition than patients ($\beta = -0.44$, 90% CI = [-0.85, -0.02]).”

10. There are key analyses that you mention in your results section and then again highlight in the discussion, but they aren’t the figures you selected. Figure 4A-D in particular seem to demonstrate that the computer did what you told it to do, and that your model parameters fit behavior well. These are useful checks, but don’t push forward the larger argument. Instead, your discussion brings up group comparisons and condition differences, but you don’t show figures of those results (with the exception of figure 2). Even then, I would include lines and color the dots to show the overall results as well as each group’s result in Figure 2C.

Reply 10:

We apologize for the lack of clarity. Fig 4 does not represent the algorithm’s behavior but the computational results from the model fitting procedure. First, we provided a visual representation of the model in Fig. 4A, which we believe greatly enhance the understanding of the model’s working, especially for less computationally-minded readers. Further, the other figures present pivotal results suggesting that 1) participants’ first choice was mostly explained by their prior expectations of their partner’s reciprocity before the interaction (Fig. 4B, which is important for subsequent analyses); 2) that participants were successfully tracking the behavior of their partners (Fig. 4C represents model-based estimates of participants’ beliefs about their partners on the x-axis, that is, ρ), which shows unbiased learning (important for our hypotheses); and 3) that participants’ trust was strictly related to our operationalization of an individual willingness to trust (Fig. 4D—again, important for the validation of our model but also for subsequent analyses and relationships as shown in Fig. 5). Importantly, the main differences that are pivotal to understand some of our previous model-agnostic findings and our subsequent analyses are depicted in Fig. 4E-F, which we explained in the manuscript as follows (pages 16-19):

“Finally, participants’ expectations of partner reciprocity as estimated by the model strongly correlated with the actual reciprocity observed ($r_{52} = .88$, $p < .00001$; Fig. 4C) and average τ parameters strongly correlated with participants’ trust in the game ($r_{52} = .93$, $p < .00001$; Fig.

4D), suggesting that our operationalization of an individual's willingness to trust strongly relates to their actual trusting behavior in the game. Further, an analysis of the policy profiles estimated by the vulnerability model shows that participants were changing their policies (i.e., probability distribution over investments) differently after positive and negative reciprocity (Fig. 4E). Specifically, the action probability (i.e., the utility-based probability of choosing a given investment) for the investment chosen by the participant at time $t - 1$ more strongly changed at time t after observing a decrease in reciprocity ($F_{(1,106)} = 13.22, p < .0005$). These policy changes were greater for lonelier ($F_{(1,106)} = 6.16, p = .0147$; Fig. 4F) and more paranoid individuals ($F_{(1,106)} = 4.10, p = .045$), corroborating the above model-agnostic findings that lonelier individuals manifested more extreme behavioral patterns (i.e., were more likely to share either everything or nothing)."

11. Figure 3A is very interesting, but I'd like it to be unpacked a bit more. I'm having trouble understanding how lonelier individuals trusted more, when the differences at high reciprocity indicates they gave a lot less. Are there group, condition, or loneliness-related differences in average reciprocity or total earnings?

Reply 11:

We tried to unpack those effects by adding additional figures (Fig. 3C-D-E) and in the text as follows (page 13):

"We observed that greater feelings of loneliness were associated with lower sensitivity to partner reciprocity, particularly in the negative condition ($\beta = -0.34, 90\% \text{ CI} = [-0.54, -0.14]$), suggesting that lonelier individuals reduced their trust in that condition more than in the other conditions (Fig. 3C). Importantly, lonely individuals' smaller increases of trust were contingent on their previous trust levels (Fig. 3D). In particular, lonelier individuals were less likely to increase their trust had they manifested lower levels of trust in the previous trial, especially in the positive condition ($\beta = 0.05, 90\% \text{ CI} = [0.01, 0.08]$) and despite having experienced positive reciprocity from the partner ($\beta = 0.47, 90\% \text{ CI} = [0.02, 0.90]$). This suggests that while less individuals might share little out of cautiousness but are ready to increase their trust upon signs of a partner's goodwill, lonelier individuals might be more suspicious, and their lower investments might more directly signal distrust."

Moreover, we explained those effects as follows (pages 13-14):

"Stronger reductions in trust in lonelier individuals could be explained by at least two mechanisms: a punitive mechanism by which lonely individuals might have wanted to punish their partner for being uncooperative and unkind; or a protective mechanism by which lonely individuals might have feared a betrayal and hence pre-emptively reduced their vulnerability to the partner. A punitive mechanism might be triggered if a trustee responds with an inadequate level of reciprocity to a participant's sign of cooperation (i.e., increase of trust). As a consequence, a decrease of trust in response to positive reciprocity could signal punitive behavior. However, as shown above, despite cases of breaches of trust, the trustees were on average reciprocating participants' increases of trust with positive reciprocity and overall, participants' trust changes positively correlated with changes in partner reciprocity ($\rho_{224} = .18, p = .0083$), suggesting that participants were seeing their trust honored by their partners and were responding by increasing their own trust.

On the contrary, a vulnerability mechanism could be triggered by the fear of experiencing breaches of trust (i.e., negative reciprocity). This fear should be greater in those interactions where there was more at stake, such as interactions with highly-reciprocating partners whose negative reciprocity meant greater losses. Indeed, interactions with highly-reciprocating partners involved higher levels of trust ($\rho_{224} = .32, p < .00001$) and since we pre-programmed all partners not to share more than half of what they received (a fairness upper bound), highly-

reciprocating partners were more likely to show negative reciprocity in future trials ($\rho_{224} = -.80, p < .00001$). This, together with lonely individuals' lower trust levels and weaker sensitivity to positive reciprocity in interactions with highly-reciprocating trustees, suggests that lonely individuals' trust was more strongly modulated by a fear of being vulnerable to a partner's betrayal. This could also explain why lonelier individuals showed patterns of trusting behaviors that more closely resembled an "all-or-nothing" strategy (Fig. 3E). They might have shared high amounts to manifest strong cooperation but, as the partner increased their level of reciprocity (Fig. 3A), they might have feared a betrayal and either completely withdrew from the interaction (i.e., sharing nothing) or reduced their trust to the lowest levels (i.e., sharing the smallest amounts) as to probe the partner's "true" reciprocity (Fig. 3E)."

Finally, we checked that our partners did not directly induce the biases we observed in our participants (page 12):

"We also checked that trustees behaved as expected and in similar fashion across groups and conditions, so as to rule out the possibility that potential biases in participants' behaviors were induced by our algorithm's different responses. Indeed, we observed that our pre-programmed partners increased their reciprocity in response to increases of participants' trust ($\beta = 7.33, 90\% \text{ CI} = [4.78, 10.02]$), and they did so in similar ways both across conditions ($\beta = -0.01, 90\% \text{ CI} = [-0.16, 0.13]$) and groups ($\beta = -0.06, 90\% \text{ CI} = [-0.48, 0.39]$). Moreover, on average, lonelier individuals did not experience lower levels of reciprocity ($r_{52} = 0.10, p = 0.495$) or earnings ($r_{52} = 0.13, p = 0.353$) from their partners."

12. In figure 5, wouldn't you want reciprocity and willingness to trust to be on the x-axis? Those are measures that happened in the game, and the ratings of fairness and trustworthiness happened after the game (if I'm understanding correctly), suggesting that they based these assessments on the behavior of the game. Regardless, these still seem to demonstrate that generally there was an interpretation that more reciprocal partners are fairer and trustworthy (i.e., the programming worked). It is interesting that those that are more willing to trust also are rate the partner as fairer (5D), going back to the argument that perhaps the player's own behavior influenced their perception of the partner. However, these figures do not that demonstrate that loneliness or paranoia, nor the impression manipulation, influenced participants' responses, which seems like a key part of the argument. You mention these things in the text, and they would be far more interesting for your figures.

Reply 12:

We thank the reviewer for this suggestion. We updated the figure accordingly. We would like to highlight that the expectation manipulation was done before the investment game and participants' impressions during their ratings of their partners were not manipulated. Moreover, here, instead of presenting the fitted lines of the effects the reviewer mentioned, we thought it might be more interesting to show plots of the relationships between real data points that could support our interpretation of the dynamics underlying those regression effects. In particular, we argued as follows (page 20):

"The formation of more negative impressions of a partner might have depended on both more negative initial expectations and lower willingness to trust, which could have led to a less successful interaction. As a consequence, partners might have reciprocated less with our participants, inducing more negative impressions. Indeed, participants' impressions positively correlated with average partner reciprocity (fairness: $r_{52} = .40, p = .003$; trustworthiness: $r_{52} = .30, p = .033$; generosity: $r_{52} = .40, p = .004$; Fig. 5B-C). Corroborating this interpretation, we observed that more positive fairness impressions were associated with more positive prior expectations of partner reciprocity ($\rho_{51} = .33, p = .017$) and a greater willingness to trust after both positive ($\rho_{51} = .45, p < .001$) and negative reciprocity ($\rho_{51} = .39, p < .004$; Fig. 5D).

Overall, these findings suggest that the formation of more negative impressions depended on both unfavorable initial expectations before the social interaction and a reduced willingness to trust a partner during the social interaction.”

13. A very minor comment about figures but be consistent with the meaning of colors in your figures.

Reply 13:

Thanks for this comment. We revised the figure and tried to be consistent with our color coding.

14. In the 5th paragraph of the discussion (“On the other...”), You mention that lonelier individuals have higher levels of trust but only in the lower levels of reciprocity. What if instead lonelier or more paranoid individuals are not considering the partner’s perspective as well? Is that a possible explanation for being more reactive to perceived slights (not receiving back as much as they anticipated)? I would think that maxing their investment and getting back half of the total (1.5 the investment) would maximize the both the player and partner’s outcomes fairly, so is it really a betrayal to not continue to return more (so the partner would be getting less)? How might theory of mind play into the game and potentially differ in lonelier or more paranoid individuals? Or do the lonelier and more paranoid players believe they should receive a greater amount than the partner?

Reply 14:

This is an interesting interpretation and we added this consideration to the Discussion. We believe that lonelier individuals might consider the partner’s perspective but might have a biased representation of the partner’s mind. Hence, even though we didn’t look at theory of mind here, we agree with the reviewer that our results provide some insights into what cognitive processes underlie theory of mind in lonelier and more paranoid individuals. In particular, lower trust in highly-reciprocating trustees suggests that lonelier individuals are distrustful of behaviors that look “too good to be true” and might interpret those behaviors as driven by malicious, strategic intent (in a paranoid-like fashion). We address this as follows (page 22):

“Importantly, these behavioral patterns also point to the fact that lonelier individuals might be biased in representing their partners’ intentional states. For instance, given lonelier individuals’ overall negative expectations of other people’s behaviors, highly-reciprocating trustees’ cooperation must have been hard to believe (in Bayesian terms: had low posterior probability) and was likely interpreted as indicative of strategic thinking and malicious intent rather than genuine kindness. This suggests potential impairments of theory of mind in lonely individuals, similar to “hypermentalization” impairments in borderline personality disorders⁶⁰ and, as questionnaire-based measures have suggested, in both loneliness⁶¹ and schizophrenia⁶².”

15. This meta-analysis of the trust game may be useful in discussing the paranoia-related findings in the context of psychosis (Prasannakumar et al., 2023, Psychological medicine) 10.1017/S0033291722002562. I’d also recommend a paper discussing a model of trustee behavior in BPD, which highlights the role of irritability in perceptions of fairness (Hula et al., 2018; PLoS Comp Biology) <https://doi.org/10.1371/journal.pcbi.1005935>. Social value orientation may be another theory that can help explain differences in expectations in outcomes (Murphy et al., 2011) <https://doi.org/10.2139/ssrn.1804189>.

Reply 15:

We thank the reviewer for suggesting this literature. To limit ourselves to the relevant literature, we discussed Prasannakumar’s paper in the Discussion as follows (page 23):

“In particular, lonely individuals might initially be prone to interact and connect with others (as shown by their overall greater willingness to trust and higher likelihood to share high amounts) but their suspiciousness of others and possible paranoid interpretations of partner reciprocity might induce malevolent intent attributions (such as excessive attributions of harm intent)^{64,65} or probing behaviors (such as reductions of trust to its lowest levels) that foster distrust and social withdrawal⁶⁶.”

16. Please add a limitations section to the discussion. It would be helpful to comment on the sample size, and in particular issues with looking at interactions of loneliness and paranoia, when it appears uncommon to get individuals who are paranoid but not lonely (although it seems like there are more people who are lonely but not paranoid).

Reply 16:

We have now added limitations and suggestions for improvements for future studies in the Discussion as follows (page 24):

“Future studies are needed to further explore the complex dynamics between expectations and behaviors in lonely individuals and overcome some of the limitations of this study. For instance, we employed a simple algorithm to generate our trustees’ behaviors, which allowed us to only investigate simple sequential effects of positive and negative reciprocity. More sophisticated algorithms to generate richer behaviors of social partners in economic games, for example, by using generative modeling approaches, might help shed light on interindividual differences in biases in social inferences and behaviors, as well as their relationships with loneliness and paranoia. Moreover, given the transdiagnostic role of loneliness, future studies need to recruit a bigger sample with a wider and more uniform distribution of both loneliness and paranoia scores, possibly by using stratified sampling, in order to gain deeper insights into the relationships between loneliness and paranoia, and their impact on social expectations and behaviors. Finally, longitudinal studies, combined with computational modeling, could provide insights into how the mechanistic processes highlighted in this study change over time and predict both a severe development of lonely states and other mental health issues such as paranoid-like reasoning patterns.”

Reply to reviewers

Reviewer #1:

As noted in my initial review, the manuscript examined the behavioural patterns and responses in loneliness and paranoia underlying social uncertainty, as captured by the investment game in a sample of 54 participants (29 participants with non-affective psychosis and 25 healthy controls). The authors are commended for providing a thorough response to my and another reviewer's feedback, particularly in clarifying the methodology and results. However, some conceptual and narrative issues remain unaddressed or inadequately addressed.

We thank the reviewer for the appreciation of our responses to the reviewers. We've now tried to address the remaining conceptual and narrative issues in the revised manuscript.

My feedback regarding the lack of a clear narrative on the functional relationship between loneliness, paranoia, and underlying social cognitive biases has not been adequately addressed. In this version of the manuscript, this relationship remained to be described superficially and imprecisely e.g., 'various mechanisms might explain these thinking patterns' (p. 4). There are instances of circular reasoning that need to be clarified and strengthened—'For instance, their elevated sensitivity to partner vulnerability might make them more cautious and suspicious about others' goodwill. Paranoid thinking might further exacerbate this heightened vulnerability by enhancing suspiciousness about others' good intentions' (p. 4). I would expect concise and direct statements that clarify whether these mechanisms are shared between loneliness and paranoia or are unique to each. This clarification is essential to provide a solid foundation for the study design and interpretation of results—such as whether a lack of trust or reciprocity is an antecedent or consequence of loneliness and/or paranoia, and whether paranoia necessarily co-occurs with loneliness and associated biased behavioral patterns and responses. Addressing this will also help shape the discussion to better highlight the theoretical implications of the analyses.

We are happy to clarify this point. First, we would like to note that our cross-sectional study cannot speak to the temporal (causal) link between maladaptive social behaviors as antecedent of loneliness and/or paranoia or the temporal relationships between loneliness and paranoia. Here, we were interested in how loneliness and paranoia impact on trust and social behavior during social interactions to shed light on predictions around the relationships between loneliness and paranoia that are following from the evolutionary theory of loneliness. Some previous work provides clues into question around causality but this is not the focus of our work. For example, previous work has shown that both loneliness and paranoia are characterized by altered social cognition and increased vigilance for social threats^{1,2} and additional empirical evidence links loneliness to paranoia via mediating cognitive-affective mechanisms³. Further, a systematic review has shown that loneliness is a risk factor for various psychopathologies, including paranoid ideation⁴, and another study suggests that early experiences and negative self-schemas (often connected to loneliness) may precede paranoia⁵. Stronger evidence for causality come from research which showed that suspiciousness toward others (i.e., paranoid thoughts) worsened in experimental manipulations of social exclusion³ and preliminary evidence has also shown that reducing loneliness leads to a decrease of paranoid symptoms⁶. We have added these considerations to the manuscript as follows (lines 57-72, pages 3-4):

"Preliminary evidence suggests that loneliness is characterized by cognitive biases that lead to overly negative expectations of others⁷. A general negativity bias may lead lonely individuals to pay more attention to threatening interactions, be more aware of negative evaluations from

others, remember more negative feedback, and form more negative expectations of others' social behaviors⁸⁻¹¹. Some evidence indicates that a negative outlook on others is more pronounced for close acquaintances than strangers¹². These negative expectations and evaluations of others in lonely individuals have been suggested to foster paranoid thinking (i.e., the inflated belief that others have malevolent intentions)¹³⁻¹⁵, which could in turn exacerbate feelings of loneliness and paranoid delusions, leading to a vicious cycle of suspiciousness and loneliness¹⁶⁻¹⁹. For example, suspiciousness towards others (i.e., paranoid thoughts) has been found to worsen over time following feelings of loneliness, with the duration of these feelings being related to decrease in trust in others^{20,21}. As preliminary questionnaire-based evidence has also shown that reducing loneliness leads to a decrease of paranoid symptoms⁶, it is plausible that feelings of loneliness might promote more paranoid thoughts^{20,22} in individuals who manifest higher levels of distrust²³⁻²⁶ and are less likely to cooperate with others^{27,28}.”

However, as can be seen, there isn't enough longitudinal evidence to clearly address the question raised by the reviewer and we cannot but speculate about those causal mechanisms. However, the fact the causal pathways that lead to loneliness and paranoia was not the focus of our work does not preclude us from investigating how loneliness and paranoia impact social behaviors (specifically, trust) in social interactions (which is the aim of this work). We have formulated our main aim more clearly as follows (lines 84-93, page 4):

“For instance, previous work has suggested a biased social learning mechanism in lonely individuals where lonelier individuals integrate more negative than positive information about others' social behaviors²⁹. This downweighing of positive information and beliefs about others could lead to similarly biased behavioral patterns. It is, thus, still unclear whether lonelier individuals form inaccurate social beliefs due to suboptimal learning or are simply reluctant to rely on positive beliefs about others¹⁸. For instance, their elevated sensitivity to partner vulnerability might be associated with more cautious and suspicious behaviors in trusting interactions. Further, paranoid thinking, which has been associated with suspiciousness about others' good intentions¹³, might be associated with heightened perception of vulnerability and, hence, lower levels of trust.”

There are also instances where coherence could be strengthened. An example is the statements “Aspects of social disconnection are pronounced in individuals with psychotic disorders and loneliness has been linked to central symptoms of psychosis, including social threat perception and anhedonia (i.e., lack of reward/pleasure from social interactions). However, the magnitude of the associations suggests that loneliness and psychosis are related, yet distinct concepts and while initial, mostly cross-sectional studies probed their interrelationship, little is known about how they impact social interactions. Hence, investigating how loneliness modulates social interactions in individuals with and without a psychotic disorder will help to address an important knowledge gap” (p. 3). Questions for clarification include: What specific “associations” are being referred to? How large is the magnitude of these associations?

They refer to the relationships between loneliness and paranoia found in previous studies. We clarified accordingly (see below).

In what way do these associations imply that “loneliness and psychosis are related”?

They are based on previous studies using different methods (e.g., correlation or regression, path analyses) and a meta-analysis, which speak to the relationships between loneliness and paranoia. We clarified accordingly (see below).

Why is it that loneliness modulates (but does not mediate) social interactions among people with and without a psychotic disorder?

To our knowledge, in the literature there is not enough evidence to speak for a mediation and this is not the focus of the current work. We have now elaborated what modulation might entail without assuming a mediating role (see below).

These statements may cause confusion for readers and blur the focus of the study, particularly regarding the role of paranoia. Clarifying these points would improve coherence and strengthen the overall argument.

We have tried to clarify those confusions by reformulating the above paragraph as follows (lines 46-56, page 3):

“Aspects of social disconnection are pronounced in individuals with psychotic disorders and loneliness has been linked to central symptoms of psychosis, including social threat perception and anhedonia (i.e., lack of reward/pleasure from social interactions)³⁰. However, previous findings indicate small-to-moderate associations between loneliness and psychotic symptoms^{31,32}, implying that while these constructs are related—particularly through overlapping social-cognitive vulnerability—they remain distinct in etiology and expression. Most available studies are cross-sectional and focus on symptom correlations rather than how loneliness shapes real-world social behaviors. Examining how loneliness modulates social interactions—such as influencing perceptions of trustworthiness, expectations of reciprocity, or responsiveness to social cues—in individuals with and without psychotic disorders may help clarify transdiagnostic mechanisms of social dysfunction and address a critical gap in the literature^{32,33}.”

Other comments are as follows:

- The use of a two-group design with a clinical sample and a healthy control sample appeared to be a merit, but it is not well justified in the Introduction.

We thank the reviewer for pointing to this. We have added a paragraph to justify the two-group design (lines 94-99, page 4):

“Here, we investigated the relationships of loneliness and paranoia with belief formation and trust in a sample of participants with and without a psychotic disorder. Including both individuals with psychotic disorders and healthy controls allows us to examine whether the associations between loneliness, paranoia, and distrust are specific to clinical psychopathology or reflect broader, transdiagnostic mechanisms, given that these processes operate along a continuum in the general population^{32,34}”

- I'm pleased to read that GPTS was used in the current study. Could the authors offer additional details of the distribution of participants by the established clinical cutoffs of GPTS, as in Freeman et al. (2021)? 'These statistics would provide support for the statement that 'paranoia should be understood as a continuum from non-clinical to clinical states' (p. 20). However, the reference cited for GPTS is the work of Freeman et al. (2021), which developed and validated the Revised-GPTS. Please clarify if GPTS or Revised GPTS was used in this study.

We are happy to clarify this point. We have reported differences in GPTS scores between the samples as follows (lines 183-186, pages 7-8):

“As expected, GPTS scores from the persecution scale (reflecting paranoia) were significantly higher in the clinical sample ($t_{52} = 2.58, p = .014$). Moreover, like for the ESM measures of paranoia (controls: $SD = 0.98$; patients: $SD = 1.46$), GPTS scores had higher variability in our clinical sample (controls: $SD = 7.61$; patients: $SD = 18.41$).”

Moreover, we used the GPTS and not the revised GPTS, as the study was run before Freeman's 2021 version. Given the differences in version, we did not provide clinical cut-offs based on this version, as suggested by the reviewer. We have updated the references accordingly. It's important to note that patients were at the point of testing clinically stable and that their inclusion was due to the fact that we were interested in the spectrum of paranoia. As previously proposed^{34,35}, paranoia should be understood as a continuum from non-clinical to clinical states and having the patients in the group enabled us to look at the wider spectrum of paranoia.

- The 'context effect' (p. 5) of $d = 0.71$ was referenced in the power calculation. However, it is not confusing what this 'context effect' refers to and how it was parameterized in the power calculation. I checked the reference cited (Fett et al., 2012) but couldn't find a clue for the effect size of 0.71.

We apologize for the confusion and are happy to clarify this point. The context effect refers to the manipulation of participants' expectations about their trustee's trustworthiness. Fett et al., 2012 reported the mean and standard deviation of trust in patients with schizophrenia ($M = 5.64, SD = 3.10$) and controls ($M = 7.73, SD = 2.70$) for this experimental manipulation in Table 3. This statistics they reported yields an effect size of $d = .71$. We clarified this in the manuscript as follows (lines 115-119, page 5):

"Our initial sample size considerations of $N = 50$ were based on the effects of a similar experimental manipulation of participants' expectations on trust in a sample of patients with schizophrenia and other psychotic disorders and a sample of healthy controls (see cooperative context of Table 3 in²⁵), which yielded an effect size of $d = .71$ calculated for an independent sample t-test, with a desired power of .80."

- The meaning of the statement 'Further, the descriptively stronger relationship in controls could suggest that in controls paranoia is more context-dependent, but less so in individuals with clinical levels of paranoia' (p. 21) is unclear. What does it mean by 'more context-dependent'? Does the stronger association in the healthy control than in patients be explained by the greater range of paranoia? The distribution of the GPTS scores would give a clue to making sense of the difference in strength of associations between the groups.

We thank the reviewer for the opportunity to address this point. We agree with the reviewer that the stronger association in the healthy controls could potentially be explained by the greater range of paranoia in that group. However, as can be seen in Fig. 2B, paranoia scores have higher variance in our *patients* group. Analyses of GPTS scores confirm the results from ESM measures (lines 183-186, pages 7-8). We revised the paragraph to make our considerations clearer as follows (lines 503-506, page 22):

"Further, the difference in the strength of the association between loneliness and paranoia (with a descriptively stronger association in controls like previous work³²) could reflect differences in how loneliness and paranoia are influenced by situational factors in the clinical and general population⁵⁹."

- Why is the case that 'the relationship between loneliness and paranoid thoughts in the control group suggests that reducing paranoid thinking could help to reduce loneliness, and that vice versa reducing loneliness may be beneficial for improving paranoid thinking in the general population' (p. 21). Does this also apply to the clinical group, although the association between loneliness and paranoia is weaker? The bidirectional relationship seems to pinpoint circular reasoning- this confusion also speaks to a lack of conceptual clarity of the relationship between loneliness and paranoia, as well as the feasibility of translational implications of treatment and interventions. The same logic issue applies to the statement 'However, it's worth noting that we do find a strong correlation between loneliness and paranoia and these two

characteristics were more strongly associated in controls than patients—again pointing to the transdiagnostic nature of loneliness and, most importantly, its relationship with mental health issues in the general population’ (p. 21). Some rewriting may help clarify the arguments presented here.

We simply meant that the relationship between loneliness and paranoid thoughts in the control group suggests that loneliness and paranoia may co-occur even in the absence of clinical psychopathology, highlighting shared mechanisms that could inform intervention and prevention in both clinical and non-clinical populations. We clarified that paragraph as follows (lines 507-518, page 22):

“Similarly, the relationship between loneliness and paranoid thoughts in the control group suggests that these two experiences may co-occur even in the absence of clinical psychopathology. This relationship and its effects on social behaviors fundamental for relationship building—such as trust—highlight shared psychological and cognitive patterns that underlie loneliness and paranoia in both patients and the general population. These findings underscore the importance of addressing maladaptive social-cognitive processes transdiagnostically, especially in preventive contexts. While the current findings cannot directly speak to the directionality of the relationship, a recent temporal analysis between loneliness and paranoia supports the hypothesis that reducing loneliness decreases paranoid thinking over time²⁰, but not vice versa. Similarly, preliminary experimental evidence also suggests that experimentally decreasing loneliness leads to a reduction of paranoia³. Further studies are needed to investigate the causal and temporal mechanisms underlying the loneliness-paranoia association.”

- I do not agree with the authors’ response that ESM was used “to have more stable estimate of participants’ loneliness and paranoia states”. Rather, its strength lies in the temporal resolution of capturing fluctuations of state loneliness and paranoia. According to Figure 2C, I guess loneliness and paranoia were operationalized as average ratings across 7 days (which is not clearly stated in the Methods section). If GPTS was used I would suggest using the GPTS total score in the analyses to better reflect the trait-like individual differences in paranoia (as formulated in the Introduction section), instead of the average of the ESM ratings. Also for the measure of loneliness, there needs to be a strong justification of how the average rating of the single, direct item of momentary loneliness reflects trait-like loneliness- it is widely discussed in the literature that trait and state loneliness take on different psychological processes and meanings.

We thank the reviewer for the opportunity to clarify this point. We agree with the reviewer that ESM and questionnaire measures may yield somewhat different insights and conclusions (we have now highlighted this in the revised manuscript). However, this does not make insights from ESM measures less valid or important. Mean ESM scores of loneliness and paranoia have the benefit to minimize memory and recall biases inherent in retrospective self-reports (like in lab-based, one-off questionnaires)³⁶⁻³⁹. For example, ESM measures of paranoia have been shown to correlate well with symptom-based measures (e.g., PANSS⁴⁰) and have been suggested to accurately assess and monitor paranoid tendencies^{41,42}. Similarly, previous work shows that variability in self-reports on loneliness makes ESM scores less biased than one-off lab-based assessments⁴³. We report these considerations together with a clearer statement of our operationalization of loneliness and paranoia in the manuscript as follows (lines 175-178, page 7):

“ESM measures were used to assess loneliness and paranoia in the flow of daily life. ESM measures have the benefit that they are less vulnerable to many sources of error that are inherent in many traditional assessment techniques (e.g., memory and recall biases), yielding more reliable measures, particularly in patients who often suffer from cognitive problems⁴¹⁻⁴³”

Further, we also agree with the reviewer that one of the strengths of the ESM is in their temporal resolution. However, investigating variability over time was not the focus of our investigation. We mentioned this in the Limitation section as something to consider in future studies (lines 624-628, page 25):

“Moreover, future studies would need to test to which degree state- and trait-like measures of loneliness and paranoia capture different dynamics in lonely individuals’ social behaviors. For example, given the heightened sensitivity to rejection in lonely individuals, some forms of social punishment (e.g., distrust but also second-party punishment) could be exacerbated in periods of high levels of momentary loneliness”

Finally, since in clinical settings, it is well established that the persecution subscale of the GPTS better capture paranoia, we have now reported the persecution scores (see above).

Reviewer #2:

I commend the authors on their efforts to revise the manuscript. I have a few additional comments.

We thank the reviewer for the appreciation of our revision efforts. We have now addressed the remaining comments.

1. How did the authors define loneliness and paranoia? It appears to be from the ESM data, but is it an average? What days were measured (before or after or around the behavioral testing?), and how frequently did participants respond to the ESM? Although it was only one question about loneliness, the references support this method, and it is benefited by having multiple assessments throughout the day. A little clarity on the data collection would be helpful, however.

We apologize for the confusion and clarified those points one by one.

For loneliness (lines 170-172, page 7):

“Loneliness was assessed for every participant based on the average score of their responses over one week to the item “I feel lonely”, as previously done¹⁵ and currently recommended by the Office of National Statistics⁴⁴.”

For paranoia (lines 172-173, page 7):

“Paranoia was assessed for every participant based on the average score of their responses over one week to the following four items previously used⁴⁵.”

For frequency and days of ESM sampling (lines 188-192, page 8):

“Questionnaires (signaled by a beep) occurred at pseudo-random moments with at least 15 min and at most 1.5 h between two consecutive throughout the day—between 8.00 am and 10.30 pm. The ESM questionnaire was completed for seven successive days and ESM items were rated on a seven-point Likert scale ranging from “not at all” (1) to “very” (7).”

2. What effect is the power analysis referencing?

We apologize for the confusion and have now clarified how the power analysis has been computed. In particular, the effect size refers to the effects of the manipulation of participants’ expectations about their trustee’s trustworthiness on their trusting behavior in a previous study. Fett et al., 2012 reported the mean and standard deviation of trust in patients with schizophrenia ($M = 5.64, SD = 3.10$) and controls ($M = 7.73, SD = 2.70$) for this experimental manipulation in Table 3. This statistics they reported yields an effect size of $d = .71$. We clarified this in the manuscript (lines 115-119, page 5).

3. At line 338 (page 13), I think a word is missing that is important for clarity: "This suggests that while less (lonely?) individuals might share little out of cautiousness but are ready to increase their trust upon signs of a partner's goodwill, lonelier individuals might be more suspicious, and their lower investments might more directly signal distrust."

We thank the reviewer for noticing this. Lonely was indeed the missing word.

4. The description of figure 3C doesn't seem to match what is in the image (it seems like trust is greater for high loneliness individuals for the neutral condition), although the text where 3C is cited is more specific. Could the figure description be a bit more specific too?

We have now made the caption more specific.

5. It seems to me that the figure description of 5A doesn't match what the figure shows. Both the figure description and the in-text reference mention positive and negative reciprocity, but that is not in the figure. I also find it confusing that the symbol tau is next to the word reciprocity, but represents a change in willingness to trust. In addition, the description uses a euphemism for willingness to trust ("manifest a stronger discrepancy"), which makes the meaning unclear. Could you please clarify here?

We apologize for the confusion. We have now clarified the caption as follows:

"The difference in willingness to trust ($\Delta \tau$) after positive and negative reciprocity was greater in lonelier participants"

6. The legend for loneliness in figure 3E doesn't match the other legends in the figure. Could this be changed to be in-line with the other sections?

We thank the reviewer for spotting this. We've changed it accordingly.

7. Please describe what tau is in the figure description if not in the figure for 4D.

We have now improved the caption as follows:

"The average of participants' τ parameters strongly correlated with their average trust"

Reply to reviewers

Reviewer #1:

Reviewer #1 (Remarks to the Author):

I appreciate the authors' time and effort in providing a thorough response to my feedback and that of another reviewer, particularly in clarifying the theoretical relationship between loneliness and paranoia, as well as in discussing the interpretations of findings and the limitations of the current study. I have two minor comments for the authors' further consideration, as outlined below:

We thank the reviewer for the appreciation of our responses.

1. In line 69 the authors described the finding of the study by Lamster et al. (2017), which is a cross-sectional study modelling the structural relationship between loneliness and paranoia. The statement that "reducing loneliness leads to a decrease in paranoid symptoms" should be softened, for example: "reducing loneliness may lead to a decrease in paranoid symptoms."

We have revised the sentence accordingly (lines 69-70, pages 3):

"As preliminary questionnaire-based evidence has also shown that reducing loneliness may lead to a decrease of paranoid symptoms"

2. In line 182, the decimal points in Spearman's correlations are missing.

We thank the reviewer for spotting this. We revised it accordingly (lines 182-183, page 7):

"Our ESM measure of paranoia was highly correlated with GPTS score for both GPTS subscales (reference: $\rho_{52} = .62$, $p < .0001$; persecution: $\rho_{52} = .74$, $p < .0001$)"

The rest of the manuscript looks good to me. This work will advance understanding of social-cognitive biases underlying loneliness and paranoia in clinical and non-clinical populations.

Thank you so much for your constructive points that helped improve our manuscript.

Reviewer #2 (Remarks to the Author):

Thank you for your responses. I have a few more comments on the new sections you added. In addition, not all of my previous comments were addressed.

We thank the reviewer for the appreciation of our revision efforts and apologies if we have not adequately replied to all previous comments. We have now addressed the remaining comments.

1. In the introduction, the word "vulnerability" gets used in what seems to be three different meanings (vulnerability to illness, a partner's vulnerability, and one's own vulnerability). In particular, I think the use of partner vulnerability was confusing in the context of the rest of the paper. Could you please clarify?

We thank the reviewer for pointing this out. We clarified the meaning of partner vulnerability (lines 90-91, page 4):

"For instance, their elevated sensitivity to being vulnerable to others might be associated with more cautious and suspicious behaviors in trusting interactions"

2. Could you please describe how often participants actually responded to the ESM? On average, did you get a 100% response rate, for instance? Were there any people excluded for not providing enough ESM responses?

We thank the reviewer for this. We clarified in the manuscript that responses varied between participants and that participants were prompted to answer questions up to 10 times a day following a reminder. We did not have any threshold to exclude participants based on their responses, given that we are interested in average estimates of loneliness and paranoia rather than temporal variation. The average amount of responses did not differ between groups. We reported these considerations in the manuscript (lines 168-169, page 7).

“Participants completed an app-based ESM questionnaire, which included questions on loneliness and paranoia up to ten times a day following a reminder”

And as follows (lines 175-176, page 7):

“ESM responses did not differ between groups for either loneliness ($t_{52} = 0.85, p = .398$) or paranoia ($t_{52} = 1.05, p = .297$)”

3. For the ESM measures, you added some discussion of the group differences and variability of the measures, but it doesn't include statistical analysis to support your claims. I also worry that if the variance is different between the two groups, this would violate the assumptions of a t-test. It's most notable for the GPTS, which you are not really using in the rest of your analyses, but if you want to include these validations of the measures, you will benefit from using a non-parametric test (assuming the group variances are, in fact, different).

We thank the reviewer for this comment. We clarified that those are descriptive differences, and, for completeness, we ran Welch's t-tests for group differences with ESM measures and GPTS, noting how results are virtually the same. We mentioned this in the manuscript as follows (lines 190-193, page 8).

“Importantly, these descriptive differences were relatively small and Welch's t-tests for unequal variance revealed virtually the same between-group results as parametric t-tests for ESM loneliness scores ($t_{52} = 2.24, p = .0297$), ESM paranoia scores ($t_{52} = 2.39, p = .0207$), and GPTS persecution scores ($t_{52} = 2.70, p = .010$)”

4. I think there is a disconnect between the story you are presenting in your figures and in the text. In particular, your discussion and conclusions highlight that the interaction of loneliness and paranoia indicate a particular sensitivity to partner reciprocity and fear of vulnerability, but this isn't presented in any of your figures directly. In general, paranoia is a small part of the story in the figures, which surprised me given the title. As it stands, the language around the interaction effect of loneliness and paranoia should be more reserved, as the sample size is low enough to make it difficult to test the effect.

We thank the reviewer for this comment. In the first round of the revision, then reviewer #2 (probably a different reviewer) asked us to remove the regression lines from the figures (including paranoia effects) and rather plot the actual data (verbatim: “I'd also like to see the actual data included, rather than just fit lines”). We followed the reviewer's suggestion, and we tried to show some of those regression effects by artificially binning the actual data by post-hoc median splits of ESM scores (which were analyzed as continuous variables). For consistency, we decided to focus on loneliness, as paranoia was taken as a potential mechanism that could explain the impact of loneliness on trust (as clarified in the Introduction). Moreover, we have added a limitation in the Limitation section related to the sample size and low sensitivity of our study's effects. Given the difficulties in recruiting big clinical samples, in the Limitation section, we suggested a conceptual replication by using subclinical levels of paranoia in bigger online samples (lines 622-626, page 25):

“In addition, our study had a relatively low sensitivity (in relation to more subtle effects revealed in the literature) and future study would need to replicate our results with a bigger sample size, potentially also using online populations (e.g., individuals with subclinical levels of paranoia), given the transdiagnostic nature of the effects of loneliness and paranoia on social behaviors observed in this study”

Finally, we have more carefully rewritten the discussion around the interaction effect of paranoia and loneliness (lines 581-582, page 24):

“suspiciousness about the partner’s true reciprocity could have reduced lonely individuals’ willingness to trust”

And (lines 593-594, page 24):

“lower willingness to trust in lonelier individuals with higher levels of paranoia could lead to overall less cooperative and trusting interactions”

5. I don’t think you addressed my concern about the terms negative and positive reciprocity in Figure 5A. What is the benefit of including these terms if you are not showing them?

We apologize for the confusion. The y-axis ($\Delta \tau$) shows the difference in willingness to trust after positive and negative reciprocity. We rephrased the caption as following:

“The difference in willingness to trust after positive and negative reciprocity (i.e., $\Delta \tau$)”

6. You did not describe tau in the caption of figure 4D.

We described it as follows:

“The average of participants’ τ parameters”